# tRNA epitranscriptomics and biased codon are linked to proteome expression in *Plasmodium falciparum*

Chee Sheng Ng[1,2,3,†], Ameya Sinha[1,2,3,†] (iD), Yaw Aniweh[2], Qianhui Nah[1], Indrakanti Ramesh Babu[3], Chen Gu[3], Yok Hian Chionh[1,4], Peter C Dedon[1,3,5,*] (iD) & Peter R Preiser[1,2,**] (iD)

## Abstract

Among components of the translational machinery, ribonucleoside modifications on tRNAs are emerging as critical regulators of cell physiology and stress response. Here, we demonstrate highly coordinated behavior of the repertoire of tRNA modifications of *Plasmodium falciparum* throughout the intra-erythrocytic developmental cycle (IDC). We observed both a synchronized increase in 22 of 28 modifications from ring to trophozoite stage, consistent with tRNA maturation during translational up-regulation, and asynchronous changes in six modifications. Quantitative analysis of ∼2,100 proteins across the IDC revealed that up- and down-regulated proteins in late but not early stages have a marked codon bias that directly correlates with parallel changes in tRNA modifications and enhanced translational efficiency. We thus propose a model in which tRNA modifications modulate the abundance of stage-specific proteins by enhancing translation efficiency of codon-biased transcripts for critical genes. These findings reveal novel epitranscriptomic and translational control mechanisms in the development and pathogenesis of *Plasmodium* parasites.

**Keywords** *Plasmodium falciparum*; quantitative proteomics; tRNA modifications

**Subject Categories** Chromatin, Epigenetics, Genomics & Functional Genomics; Microbiology, Virology & Host Pathogen Interaction; RNA Biology

**Mol Syst Biol. (2018) 14: e8009**

## Introduction

Malaria persists as a major global disease, with the majority of deaths caused by *Plasmodium falciparum*. The asexual intra-erythrocytic developmental cycle (IDC) of *P. falciparum* is responsible for the major clinical manifestations of malaria. Consistent with the requirement for coordinated gene expression in the parasite developmental life cycle, the transcriptional changes over the *P. falciparum* IDC are tightly regulated and exhibit a "just-in-time" expression pattern wherein transcription correlates with the physiological demands of specific time points in the developmental cycle (Bozdech et al, 2003; Le Roch et al, 2003; Otto et al, 2010). The mechanisms controlling this transcriptional regulation are poorly understood, and relatively few transcription factors have been identified for this dynamic regulation (Coulson et al, 2004; Balaji et al, 2005), though alterations in chromatin structure may play a role (Ponts et al, 2010, 2011). There is emerging recognition of new mechanisms of gene expression in *Plasmodium* involving multiple levels of control. Studies comparing transcriptional and translational profiles in the IDC reveal poor correlation and delay of up to 18 h in transcript and protein levels (Le Roch et al, 2004; Foth et al, 2008, 2011; Bunnik et al, 2013), which suggests important roles for post-transcriptional and translational regulation of gene expression in the IDC of *P. falciparum* (Vembar et al, 2016). Proposed mechanisms for post-transcriptional control in *P. falciparum* include mRNA processing and degradation (Brengues et al, 2005; Newbury, 2006; Deitsch et al, 2007; Parker & Sheth, 2007; Sims et al, 2007; Yamasaki & Anderson, 2008; Horrocks et al, 2009), translational repression (Hall et al, 2005; Mair et al, 2006; Parker & Sheth, 2007; Shaw et al, 2007; Abaza & Gebauer, 2008), translational regulation by untranslated regions (UTR) in mRNA (Hasenkamp et al, 2013; Brancucci et al, 2014; Cui et al, 2015), and natural (endogenous) anti-sense transcripts (Patankar et al, 2001; Kyes et al, 2002; Gunasekera et al, 2004; Militello et al, 2005, 2008; Lu et al, 2007). However, these

1  Antimicrobial Resistance Interdisciplinary Research Group, Singapore-MIT Alliance for Research and Technology, Singapore City, Singapore
2  School of Biological Sciences, Nanyang Technological University, Singapore City, Singapore
3  Department of Biological Engineering, Massachusetts Institute of Technology, Cambridge, MA, USA
4  Department of Microbiology and Immunology Programme, Yong Loo Lin School of Medicine, National University of Singapore, Singapore City, Singapore
5  Center for Environmental Health Sciences, Massachusetts Institute of Technology, Cambridge, MA, USA
   *Corresponding author. Tel: +1 617 253 8017; E-mail: pcdedon@mit.edu
   **Corresponding author. Tel: +65 6316 2822; E-mail: prpreiser@ntu.edu.sg
   †These authors contributed equally to this work

gene-specific mechanisms which have local effects do not explain a more global systematic scale mechanism involved in the control of *Plasmodium* gene expression at the level of translation.

Among candidate translational control mechanisms operant in other eukaryotes (Meyer & Jaffrey, 2014; O'Connell, 2015), the modified ribonucleosides of the epitranscriptome have not been extensively explored in *Plasmodium* parasites in spite of comparative genomics analyses of tRNA-modifying enzymes (Sawhney *et al*, 2015). We have described a translational regulatory mechanism in eukaryotes and prokaryotes involving growth- and stress-specific reprogramming of post-transcriptional modifications on tRNA linked to selective translation of codon-biased gene families (Chan *et al*, 2010, 2012, 2015; Dedon & Begley, 2014; Su *et al*, 2014; Deng *et al*, 2015; Endres *et al*, 2015; Chionh *et al*, 2016; Doyle *et al*, 2016). The nuclear genome of *P. falciparum* contains a total of 46 tRNA genes encoding 45 tRNA isoacceptors (two different genes encode the initiator and elongator tRNA$^{Met}$; Gardner *et al*, 2002). While all nuclear-encoded tRNAs in *P. falciparum* are similar to other eukaryotic tRNAs in terms of semi-conserved sequences and structure (Preiser *et al*, 1995; Pütz *et al*, 2010), it is striking that *P. falciparum* has the smallest set of tRNA genes for a eukaryotic cell, whereby there exists only one gene copy per tRNA isoacceptor (Gardner *et al*, 2002) for the nuclear genome. This unique characteristic of *P. falciparum* highlights the potential importance of epitranscriptomic modifications in a complex regulatory network that accurately decodes 61 codons by 45 cytoplasmic tRNA isoacceptors.

Here, we present system-level proteomic, genomic, and epitranscriptomic analyses in *P. falciparum* that provide new insights into tRNA maturation and reveal a coordinated system of tRNA modification reprogramming coupled to codon-biased translation to fine-tune parasite protein needs across the IDC. These observations have implications for the pathobiology of other eukaryotic parasites and identifying new targets for development of antimalarial drugs and infection biomarkers.

## Results

### The spectrum of tRNA modifications in *P. falciparum* is similar to other eukaryotes

Given the nearly complete lack of information about RNA modifications in malaria parasites, we first identified the complete spectrum of modified ribonucleosides in tRNA from *P. falciparum*. As outlined in the workflow in Appendix Fig S1A, care was taken to selectively rupture RBC membranes with saponin to purify parasites (Gruenberg & Sherman, 1983) and then extract total RNA. Bioanalyzer analysis (Appendix Fig S1B) revealed tRNA and 5S, 5.8S, 18S, and 28S rRNAs from *P. falciparum* and the apicoplast organelle, with minor amounts of human 18S and 28S rRNA, which confirmed limited host cell RNA contamination. Kit-based small RNA purification (Appendix Fig S1C and D) followed by size exclusion chromatography provided a relatively pure tRNA population (Appendix Fig S1E; Chionh *et al*, 2013; Cai *et al*, 2015). Following enzymatic hydrolysis and chromatography-coupled mass spectrometry (LC-MS; Cai *et al*, 2015), we identified 28 modified ribonucleosides as the initial basis set (Fig 1A, Appendix Table S1), of which

22 were validated against synthetic standards and six were identified by collision-induced dissociation (CID) fragmentation patterns and high mass accuracy MS.

Given the lack of information about parasite RNA modifications, we compared the set of 28 to those previously identified in Eukarya, Bacteria, and Archaea (Fig 1B; Cantara *et al*, 2011; Jackman & Alfonzo, 2013; Machnicka *et al*, 2013), and found the majority common to all domains of life and all but one (m$^4$Cm) found in Eukarya. *P. falciparum* have both apicoplast and mitochondria organelle, which contain their own genomes and are translationally active, though the mitochondria lack tRNA and tRNA-modifying genes (Gardner *et al*, 2002; Jackson *et al*, 2011). The presence of organellar tRNA modifications of bacterial origin—such as f$^5$C, f$^5$Cm, k$^2$C, ho$^5$U, tm$^5$U, tm$^5$s$^2$U, and cmnm$^5$U (box in Fig 1B)— might be expected in our total tRNA population and would thus most likely be derived from the apicoplast. The relatively minor contribution of total apicoplast RNA (rRNAs 0.5–2% of nuclear-encoded rRNAs; Appendix Fig S1B) may have resulted in most modification levels below the limit of detection of our LC-MS method. m$^4$Cm-modified tRNA could thus be either cytosolic or apicoplast tRNAs.

Focusing our comparative analysis of tRNA modifications in the eukaryotes (Fig 1C), we observed that two of these modifications (ms$^2$t$^6$A, m$^3$U) were not shared by *S. cerevisiae* and *Homo sapiens* but were present in the parasites. The presence of three modifications not shared with the human host (m$^4$Cm, ms$^2$t$^6$A, m$^3$U) raises the potential for targeting their biosynthetic pathways for drug development. The comparison also revealed that the three organisms all share the same repertoire of eukaryotic wobble modifications ncm$^5$U, mcm$^5$U, mcm$^5$s$^2$U, Ψ, s$^2$U, Gm, Cm, I, and m$^5$C (Fig 1C). The one exception to this catalogue was ncm$^5$Um (Agris *et al*, 2007; Weixlbaumer *et al*, 2007; Agris, 2008). This conservation facilitates the localization of specific modifications to positions in tRNA isoacceptors in studies linking IDC stage-specific modification changes to codon-biased translation given the importance of wobble modifications in modulating codon–anticodon interactions.

### Parasite tRNA modifications vary across the IDC

Having identified a set of 28 tRNA modifications, we next quantified changes in the levels of each modification across the 48-h *P. falciparum* IDC. Here, we isolated tRNA from highly synchronized (> 80% synchrony) cultures at different time points spanning the entire IDC: 8 h—early ring stage; 16 h—middle ring stage; 24 h—the ring-to-trophozoite transition; 32 h—trophozoite stage; 40 h—the trophozoite-to-schizont transition; and 46 h—mature schizont stage (Fig 2A). As noted earlier, measures were taken to minimize contamination with RBC RNA and to prepare highly purified tRNA at each time point (Appendix Fig S1F and G). Following LC-MS/MS analysis of the set of tRNA modifications in each sample, hierarchical clustering analysis of fold-change values for each modification (relative to a time-course average) distinguished the three major IDC stages of ring, trophozoite and schizont, as well as the sequential steps in the time-course (Fig 2A). The heat map also reveals three patterns of up- and down-regulation of the 28 modifications. The major pattern involves simultaneous and significant increases in the levels of 22 of 28 modifications at 24–32 h of the IDC, as

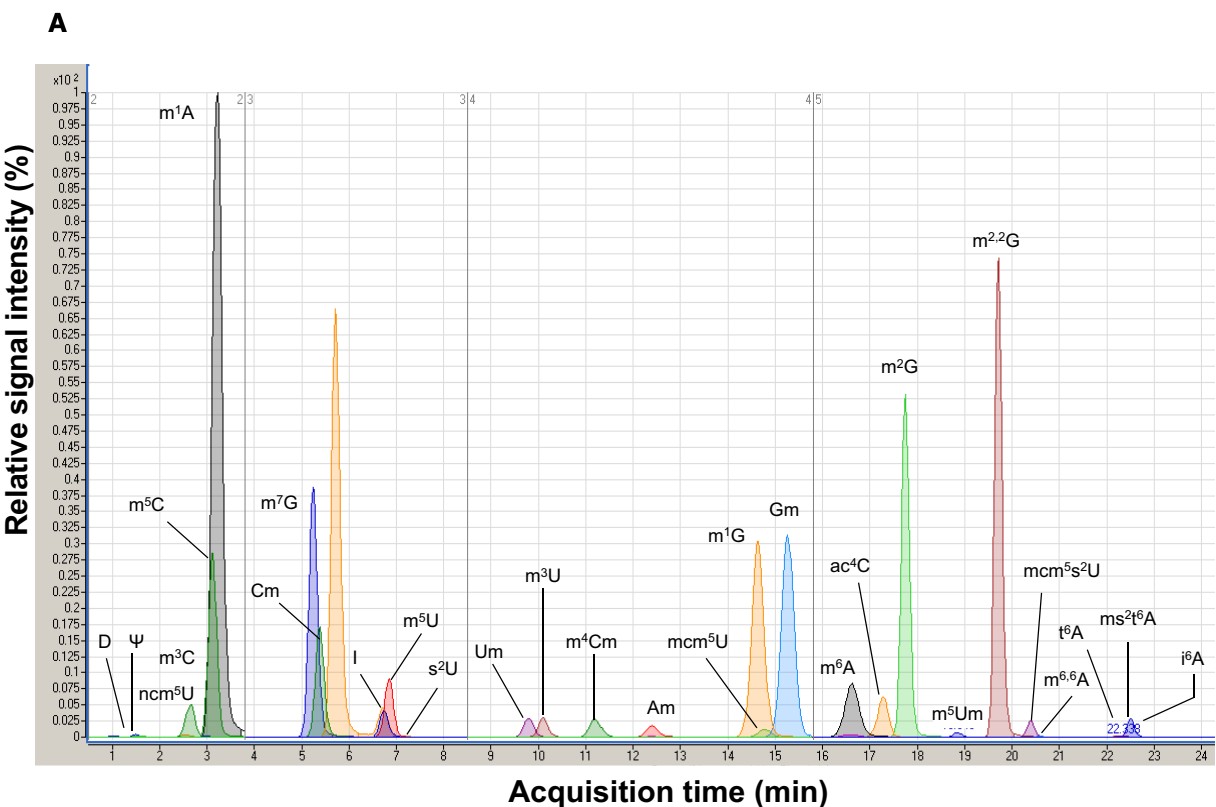

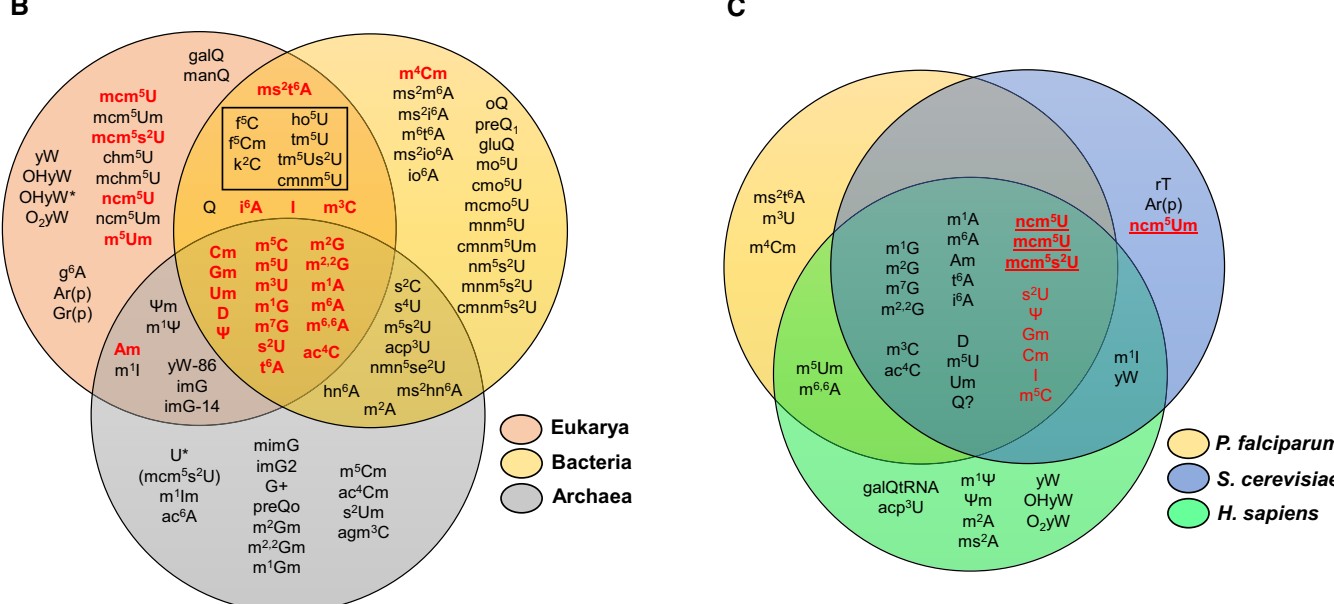

**Figure 1. *Plasmodium falciparum* tRNA modifications.**

A   LC-MS/MS extracted ion chromatograms of 28 modified ribonucleosides analyzed in parasite tRNA. Signal intensities of the modifications are scaled to that for m$^1$A, the ribonucleoside with the highest MS signal.

B   Comparison of *P. falciparum* tRNA modifications (red) to those in all three kingdoms of life (Grosjean & Benne, 1998; Jackman & Alfonzo, 2013). Modifications in the box are those found in eukaryotic cellular organelles, and the assignment of ms$^2$t$^6$A to both eukarya and bacteria is supported by published work (Arragain *et al*, 2010). The figure has been adapted from Grosjean and Benne (1998) and Jackman and Alfonzo (2013).

C   Comparison of tRNA modification profiles among *P. falciparum*, *S. cerevisiae*, and *H. sapiens*. tRNA wobble modifications are highlighted in red, with wobble-specific modifications bolded and underlined. The assignments of tRNA modifications to *H. sapiens* and *S. cerevisiae* are based on published work (Jühling *et al*, 2009; Chan *et al*, 2010; Phizicky & Hopper, 2010; Cantara *et al*, 2011; Machnicka *et al*, 2013; Kellner *et al*, 2014; Su *et al*, 2014).

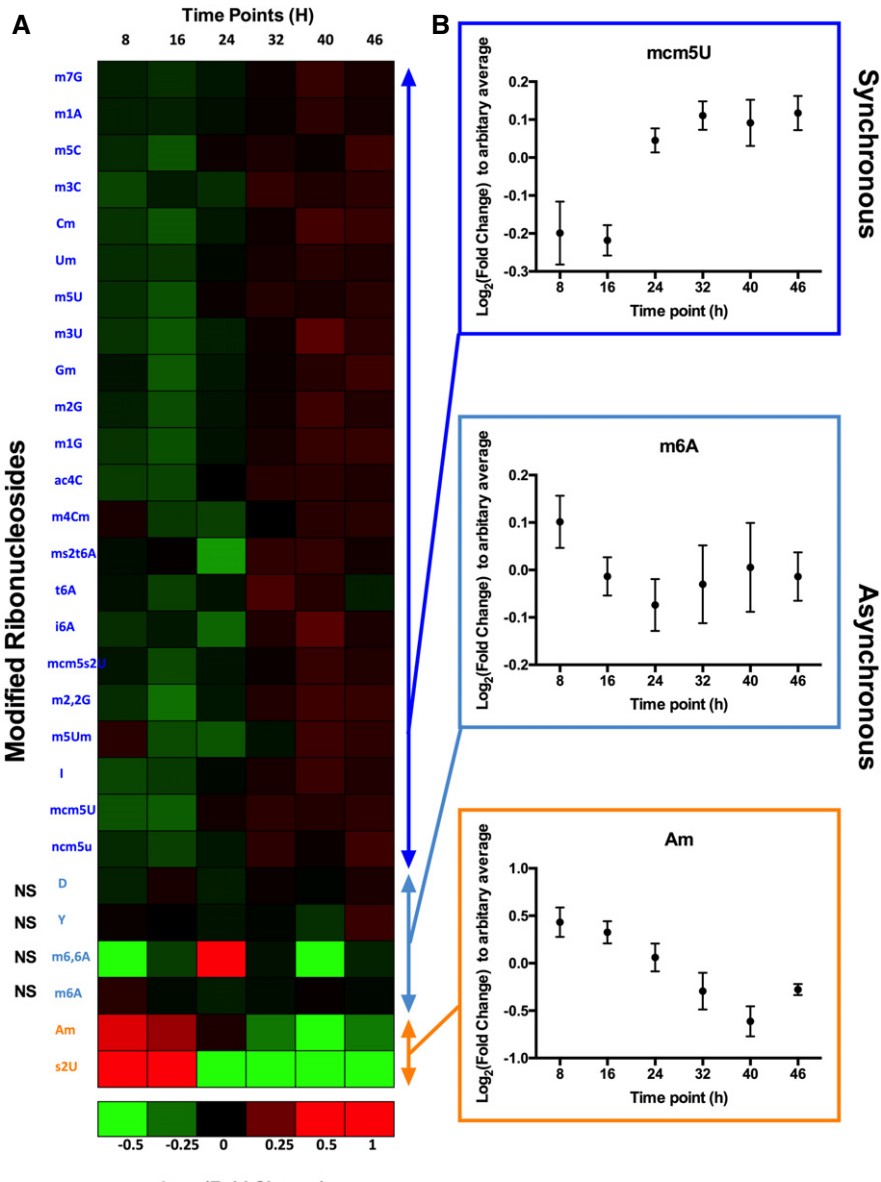

**Figure 2. Stage-specific reprogramming of 28 tRNA modifications across the *P. falciparum* IDC.**

A Changes in the relative quantities of individual modified ribonucleosides were quantified by LC-MS/MS in total tRNA extracted from parasites across the IDC time-course. The average fold-change values (relative to an arbitrary average control) were subjected to hierarchical clustering analysis (log$_2$-transformed data). The scale of the heat map has saturated the fold-change values of some modifications. Refer to Appendix Table S6.

B Three groups of modifications show differential regulation during the course of IDC. The majority (22) increase synchronously across the IDC (blue), while four show asynchronous or irregular behavior (purple) and two (orange) are uniquely up-regulated in the ring stage. Log$_2$Fold-change values plotted in the three graphs represent mean ± SEM (*N* = 3). Differences between the highest value (denoted as peak) and the lowest value (denoted as trough) were subjected to a two-tailed Student's *t*-test: NS, not significant.

illustrated graphically for mcm$^5$U in Fig 2B. The second pattern shows the opposite behavior for Am and s$^2$U, with decreased levels across the IDC (Fig 2B), while a third set of modifications (D, Ψ, m$^6$A, and m$^{6,6}$A) showed either statistically insignificant changes or discontinuous patterns of change across the IDC (Fig 2B). These three patterns suggest a multifaceted program of tRNA modification changes during parasite development and raised the possibility of a direct link to translation of stage-specific proteins.

## Stage-specific protein expression across of IDC

To test any possible link between tRNA modification changes and regulation of gene expression at the level of translation, we next performed iTRAQ-based quantitative proteomics across the IDC, using an optimized protein extraction and data analysis protocol (Appendix Fig S2A). Analysis of LC-QTOF data for three biological replicates and matching to *H. sapiens* and *P. falciparum* protein

databases allowed us to map 38,325 identified spectra (1,022,590 total spectra; peptide threshold < 1% FDR) onto 2,119 *P. falciparum* proteins (protein threshold > 99% confidence, minimum two peptides; Appendix Fig S2A), which represents 40% of the predicted 5,300 genomically encoded proteins (Gardner *et al*, 2002). Of these proteins, 2,100 were consistently quantified in all time points with good Gene Ontology (GO) representation (Appendix Fig S2A–E, Appendix Table S7). Compared to the other datasets that exist, our dataset provides for sufficient quantitative data across a time resolution that is compatible with our analysis (Florens *et al*, 2002; Foth *et al*, 2008, 2011). Proteins with > 1.2-fold and < 0.83-fold changes were defined as up- and down-regulated, respectively. As shown in Fig 3, Malaria Parasite Metabolic (MPM) and GO pathway enrichment organizes annotated gene products and provides functional context (Ginsburg & Abdel-Haleem, 2016). This analysis performed for the top 100 up-regulated proteins at each time point revealed functionally appropriate differences in protein expression at each IDC stage. For example, the proteins responsible for cytoadherence of trophozoites and schizonts and invasion in the merozoite stage are up-regulated during the trophozoite (32–40 h) and schizont (40–46 h) stages, respectively. Similarly, up-regulation of metabolism proteins occurs late in the ring stage (24 h) preceding the higher-intensity

maturation process of the trophozoite stage. The ramp-up of translational machinery at the trophozoite–schizont border is also consistent with the preceding bulk maturation of tRNA in terms of modifications (Fig 2A), while any link between tRNA modification reprogramming and codon-biased translation requires quantitative understanding of codon usage patterns.

## An alternative genetic code of synonymous codons correlates with protein up- or down-regulation in late stages of the IDC

Having established stage-specific protein regulation in the *Plasmodium* IDC, we next used a codon-counting algorithm to quantify codon usage patterns in the 100 most highly up- and down-regulated proteins at each time point (Appendix Table S9A and B). The goal was to identify significant codon usage biases that are hypothesized to correlate with tRNA wobble modification changes. Here, we first applied principal component analysis (PCA) to investigate the relationships among the stage-specific proteome changes and the codon usage patterns of genes for up-regulated proteins. The scores plot in Fig 4A shows that proteins up-regulated at 40 and 46 h are distinguished mainly in PC1 from proteins at earlier time points (8, 16, 24, 32 h) with respect to their synonymous codon usage. The distinctions among proteins subject to stage-specific up- and down-regulation are

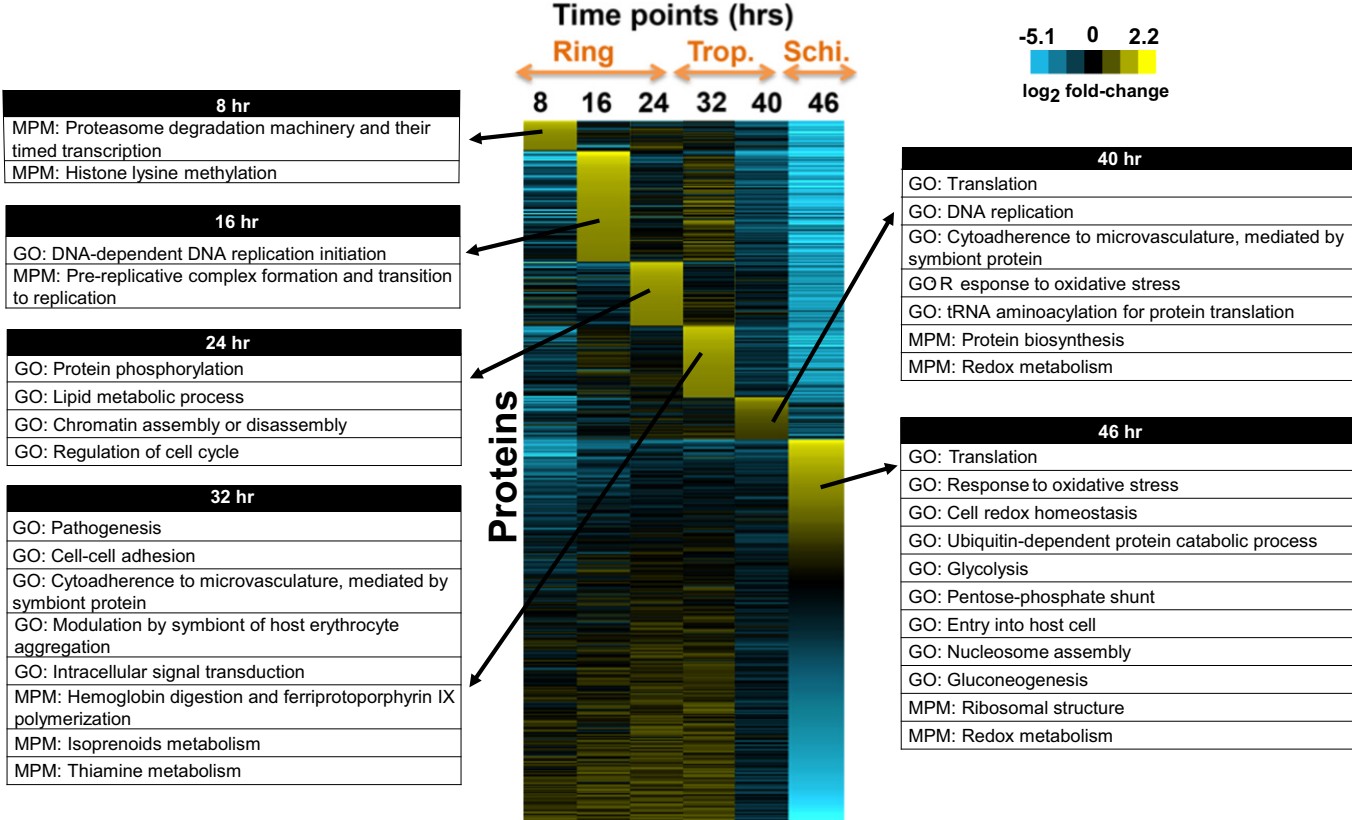

**Figure 3. Stage-specific up- and down-regulation of *P. falciparum* proteins across the IDC.**
iTRAQ proteomics analysis (Appendix Fig S2) resulted in 2,100 proteins consistently quantified at every point across the IDC. The heat map shows hierarchical clustering analysis of fold-changes (log$_2$-transformed) in the levels of the proteins relative to an arbitrary average across the IDC. The 100 most up-regulated proteins at each time point were subjected to pathway enrichment analysis based on GO and MPM pathways annotated for *P. falciparum* 3D7. The significantly ($P < 0.05$) enriched pathways among the up-regulated proteins at each time point are summarized in tables for each time point.

more clearly illustrated in the scores plot from a partial least squares (PLS) regression analysis that focuses on late-stage (40–46 h) proteins in Fig 4B (red and green symbols, respectively). Here, there is a sharp distinction between up- and down-regulated proteins, with the

loadings plot (Fig 4C) revealing a strong association of codons for 13 amino acids with the up-regulated proteins: Arg$^{AGA}$, Gly$^{GGA}$, Glu$^{GAA}$, Pro$^{CCA}$, Ser$^{TCA}$, Val$^{GTT}$, Leu$^{TTA}$, Ala$^{GCT}$, Ile$^{ATC}$, Ile$^{ATT}$, Asn$^{AAC}$, His$^{CAC}$, Tyr$^{TAC}$, and Cys$^{TGC}$ (Fig 4C, red), and a significant anti-correlation of

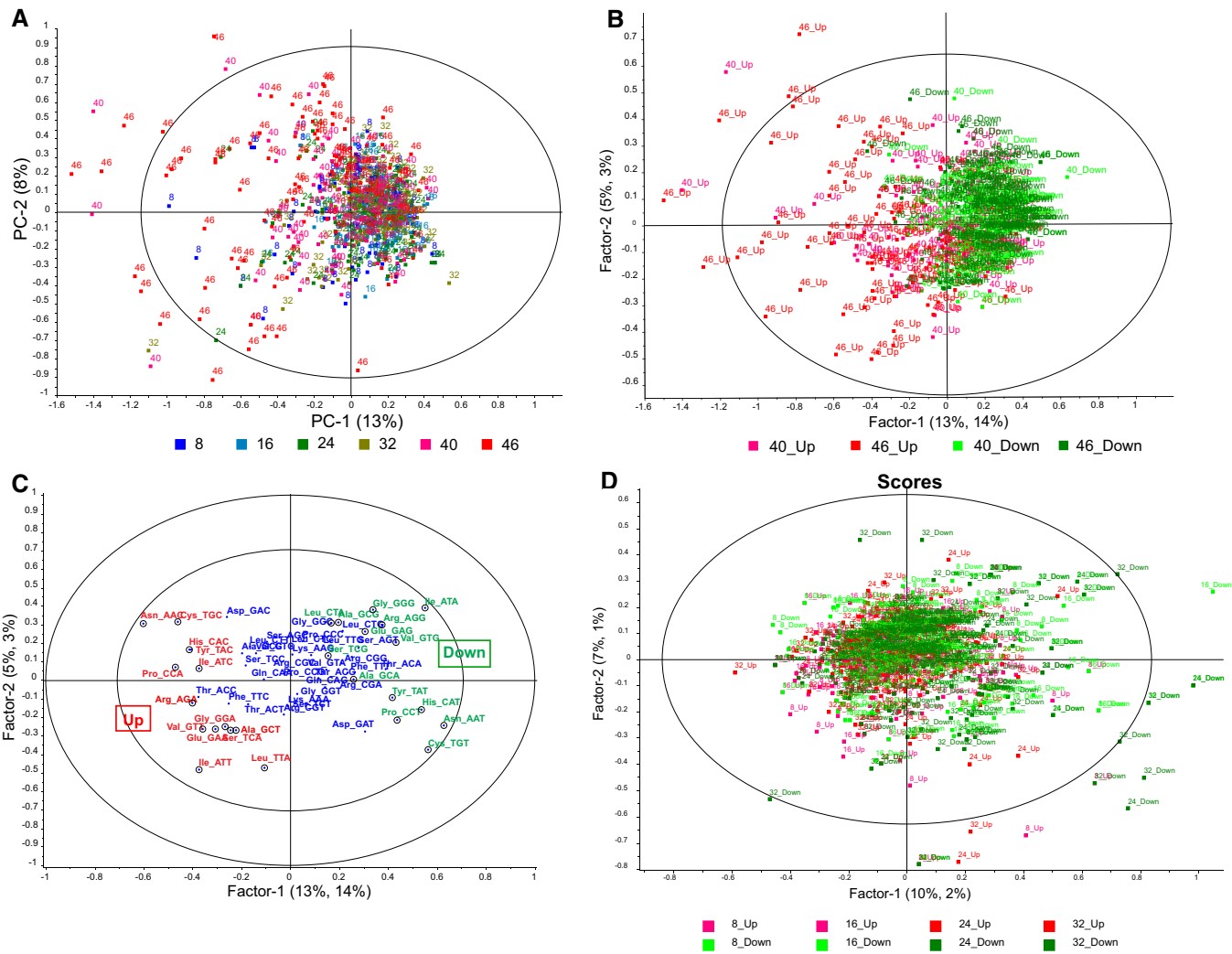

**Figure 4. A code of synonymous codons regulates translation during late stages of the *P. falciparum* IDC.**

Here, we tested the hypothesis that highly up- and down-regulated proteins at late stages of the IDC are derived from genes with biased use of synonymous codons that match the stage-specific tRNA modification changes. The 100 most up- and down-regulated proteins at each stage of the IDC were analyzed for codon usage patterns. The codon usage percentages in each gene were used to prepare a data matrix for principal component analysis (PCA) along with the fold-change up- and down-regulation for each protein at each time point.

A   PCA of codon usage percentages of up-regulated proteins at each time point. The scores plot shows that proteins for 40 and 46 h (red labels) are distinguished from the other time points (green and blue labels) mainly in PC1. The color legend for IDC time points is shown below the *x*-axis.

B, C   The proteomic data for TP40 and TP46 were also analyzed by partial least squares (PLS) regression for codon usage percentages in up- or down-regulated proteins. (B) The scores plot shows a clear distinction between up- and down-regulated proteins at both late-stage time points (40_Up, up-regulated at TP40; 46_Up, up-regulated at TP46; 40_Down, down-regulated at TP40; 46_Down, down-regulated at TP46). (C) The corresponding *x, y* correlation loadings plot for the analysis in panel (C) shows the codons contributing most strongly to this separation of up- and down-regulated proteins. The circled codons are the significantly contributing codons determined by the PLS analysis. For ease of visualization, red denotes synonymous codons most over-represented in the up-regulated proteins (TP40, TP46), while green denotes the corresponding synonymous codons that turn out to be enriched in down-regulated proteins at TP40 and TP46; blue represents the codons with the lowest degree of enrichment in either up- or down-regulated proteins.

D   PLS regression analysis of codon usage in genes for up- or down-regulated proteins at 8, 16, 24, and 32 h of the IDC (TP8-32). The scores show no significant distinctions among up- and down-regulation on the basis of codon usage. The label key is similar to that in panel (B).

Data information: Ellipses in the scores plots represent the Hoteling *T*$^2$ limit, *P* < 0.01 (*F*-test), while the outer and inner ellipses in the loadings plots indicate 100 and 50% explained variance, respectively.

these codons with their synonymous counterparts enriched in down-regulated proteins: Arg$^{AGG}$, Gly$^{GGT}$, Glu$^{GAG}$, Pro$^{CCT}$, Ser$^{TGC}$, Val$^{GTG}$, Leu$^{CTA}$, Ala$^{GCA}$, Ile$^{ATA}$, Asn$^{AAT}$, His$^{CAT}$, Tyr$^{TAT}$, and Cys$^{TGT}$ (Fig 4C, green). The most striking feature here is that these pairs of synonymous codons are differentially enriched in genes for proteins undergoing opposite changes in late-stage parasites (40 and 46 h): Codons over-represented in up-regulated proteins are under-represented in the down-regulated proteins, and *vice versa* (Fig 4C). An important point here is that these codon associations do not show any correlation with the traditional indices of average codon usage or codon "optimality" (i.e., genome-wide average use; Appendix Table S2; Sharp & Li, 1987). Finally, the multivariate statistical analyses shown in Fig 4D reveal a clear lack of codon over- or under-usage in up- and down-regulated proteins at earlier time points (8–32 h). These results point to an alternative genetic code of biased use of synonymous codons in families of stage-critical genes in the late phase of parasite development. This raises the question of the link between stage-specific tRNA reprogramming and stage-specific codon biases.

### IDC stage-specific tRNA reprogramming is correlated with codon-biased translation

An initial link between stage-specific tRNA modification reprogramming and codon-biased translation appeared in the observation that, of the 13 codons associated with up-regulated proteins in the late IDC (Fig 4C), nine are read by tRNA isoacceptors with modified wobble ribonucleotides (Agris *et al*, 2007; Johansson *et al*, 2008; Grosjean *et al*, 2010). These include tRNA$^{Arg(\underline{mcm5UCU})}$, tRNA$^{Gly(\underline{mcm5UCC})}$, tRNA$^{Glu(\underline{mcm5s2UUC})}$, tRNA$^{Pro(\underline{ncm5UGG})}$, tRNA$^{Ser(\underline{ncm5UGA})}$, tRNA$^{Val(IAC)}$, tRNA$^{Leu(\underline{ncm5UmAA})}$, tRNA$^{Ala(IGC)}$, and tRNA$^{Ile(IAU)}$ (Appendix Table S3). These wobble modifications were also observed to be highly up-regulated in tRNA at 40 and 46 h (Fig 2A), except for ncm$^5$Um that was not identified in this study, and all enhance reading of their cognate codons (Agris *et al*, 2007; Johansson *et al*, 2008; Grosjean *et al*, 2010). We validated the presence of wobble mcm$^5$s$^2$U and mcm$^5$U in tRNA isoacceptors for glutamine, glycine, and glutamic acid using an RNase-coupled mass spectrometry approach (Fig 5A–F, Appendix Fig S3 and Appendix Table S4; Chan *et al*, 2012; Cai *et al*, 2015; Chionh *et al*, 2016). This enabled the assignment of wobble mcm$^5$s$^2$U to tRNA isoacceptors for glutamic acid and glutamine (Fig 5) and wobble mcm$^5$U in the glycine tRNA acceptor (Appendix Fig S3). Traditionally, the effects of codon usage bias on translation have been attributed to the copy numbers of available tRNAs. A preliminary quantification of these tRNA copy numbers (Appendix Fig S4) from our mass spec data corresponds with the trends of the data provided by the microarray quantitation of the *Plasmodium* lifecycle (Rovira-Graells *et al*, 2012). These depict a relatively unchanging pool of tRNA isoacceptors for glutamine, glutamic acid, and glycine suggesting that there is more than conventional wisdom has to offer. Figure 5G–I presents the percentage of the wobble U-modified tRNA fragment for all of the three isoacceptors as the parasite progresses in its lifecycle from ring to the trophozoite stage. From the 10% of the population of glutamine (CAA) and glycine (GGA) that was modified in the ring stage, 33% of both isoacceptors is modified in the trophozoite stage. A more drastic increase is observed in the glutamic acid (GAA) isoacceptor whereby 67% of the tRNA is modified (Appendix Table S6D and E). Overall, this corresponds

with the trends presented in Fig 2A which assays for a global change in modified nucleosides. There is a limitation to this approach which must be pointed out. The summation for 100% of the specific tRNA isoacceptor in the population is taken only considering one modification in the anticodon loop. The permutation of possible modifications in the anticodon hydrolyzed fragmentation is an exponential task and would require an orthogonal approach.

On the other hand, the cognate tRNA isoacceptors that decode the 12 codons enriched in down-regulated proteins (Fig 4C), in contrast to those associated with up-regulated proteins, are not known to be modified at the wobble position, except for tRNA$^{Ala(\underline{ncm5UGC})}$ and tRNA$^{Ile(\Psi A\Psi)}$ (Appendix Tables S3 and S5; Agris *et al*, 2007; Johansson *et al*, 2008; Grosjean *et al*, 2010). These correlations between IDC stage-specific wobble tRNA reprogramming and codon-biased translational up- and down-regulation suggest that a coordinated selective translation mechanism plays a role in shaping the late-stage specific proteome.

### Translational efficiency accounts for codon-biased up-regulation of proteins in the late stages of the IDC

The correlative link between tRNA modification reprogramming and up-regulation of proteins from codon-biased transcripts in late-stage IDC suggests the possibility that this coordinated activity could increase the efficiency of translation of stage-critical proteins. If this is the case, then the abundance of late-stage proteins (40, 46 h) should increase independently of mRNA levels but in parallel with the altered levels of associated tRNA wobble modifications. This model was tested by quantifying an arbitrary index of translational efficiency (TE) defined as the ratio of the fold-change value for a protein to the fold-change value for the corresponding mRNA at a specific time point (Appendix Table S8). This represents an estimate of translational output per mRNA copy and is similar to other published TE indices (Schwanhausser *et al*, 2011). We chose RNA-seq data from Otto *et al* (2010) as it not only matches the time points of our proteomic data and parasite strain but also measures steady-state RNA levels, which is more relevant for steady-state translational activity as compared to other datasets which may have better in-depth coverage (Caro *et al*, 2014; Lu *et al*, 2017). This analysis, however, fails to account for protein degradation which most likely contributes to the steady-state level of proteins. Following normalization of proteomics and published RNA-seq (Otto *et al*, 2010) data to account for different scales, TE values for the 100 most highly up-regulated proteins at late trophozoite (40 h) and schizont (46 h) stages were calculated and compared to TE values for the same genes at other IDC time points (one-way ANOVA). As shown in Fig 6A and B, the TE values for proteins up-regulated at 40 and 46 h are higher than TE values at other time points. Furthermore, the magnitude of the TE values at 46 h is higher than those at 40 h, which is consistent with increasing translational efficiency as the parasites mature.

These results suggest the existence of a mechanism that enhances the translational efficiency of stage-critical proteins. One possible explanation is that the most efficiently translated proteins are those that possess codon biases linked to up- and down-regulated proteins (Fig 4). To test this idea, we identified proteins with the highest and lowest translational efficiency and then assessed

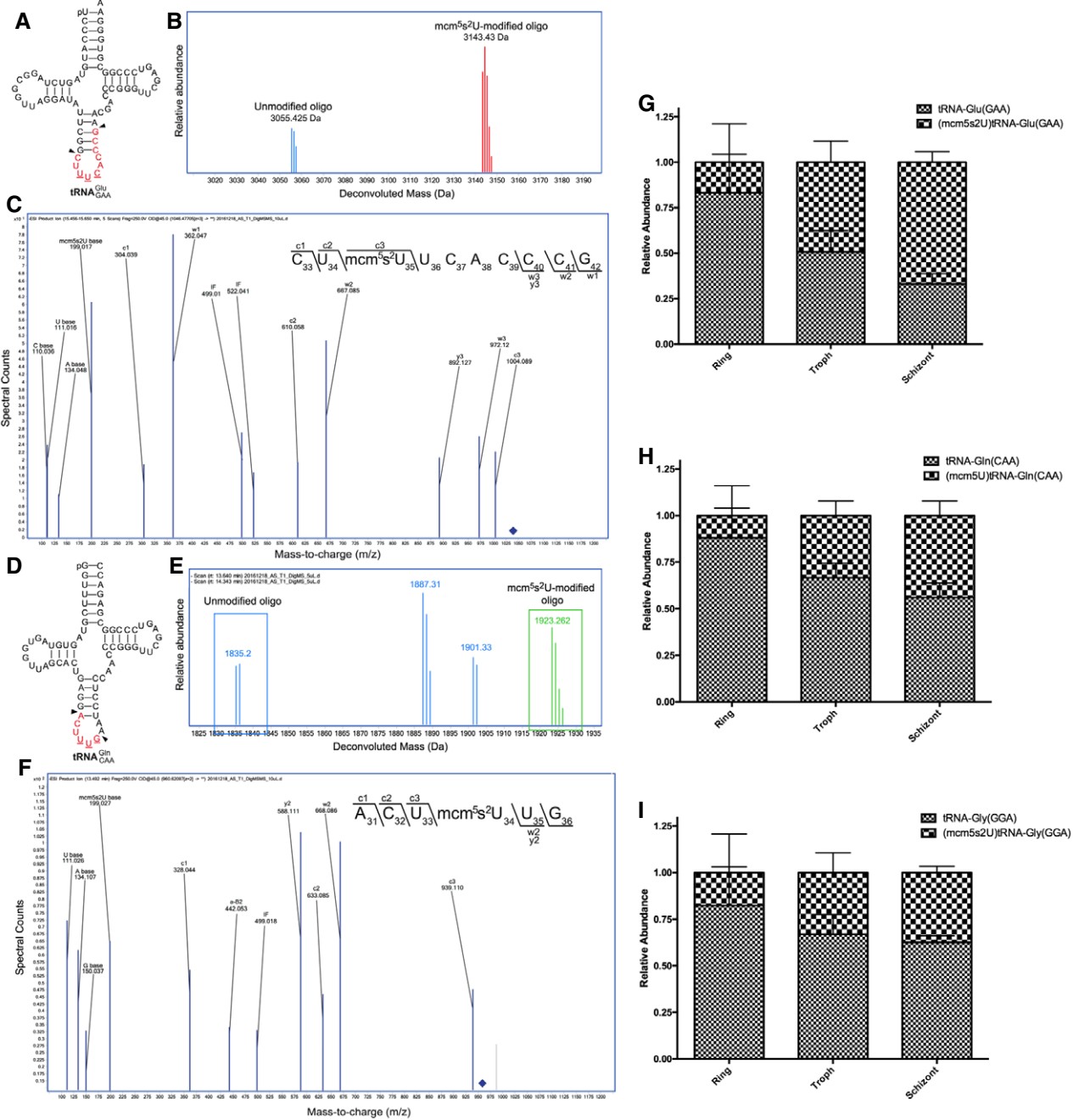

**Figure 5.** Up-regulated mcm⁵s²U maps to wobble positions in Glu-tRNA$^{GAA}$ and Gln-tRNA$^{CAA}$, whose codons are enriched in up-regulated proteins.

A–I (A, D) Sequences of *Plasmodium falciparum* Glu-tRNA$^{GAA}$ (A) and Gln-tRNA$^{CAA}$ (D). The arrows indicate T1 digestion sites. (B, E) LC-MS/MS analysis was performed on T1-digested tRNA and deconvolution yielded precise masses of the unmodified (3,055.425 Da) and modified (3,143.43 Da) oligos for Glu-tRNA$^{GAA}$ (B) and unmodified (1,835.20 Da) and modified (1,923.262 Da) oligos for Gln-tRNA$^{CAA}$ (E). (C, F) Collision-induced dissociation of the parent molecular ions for *m/z* $1,046.477^{-3}$ from Glu-tRNA$^{GAA}$ and *m/z* 1,923.262 from Gln-tRNA$^{CAA}$ produced expected −c, −w, and −y ions for the sequences CU[mcm⁵s²U]UCACCCG and CC[mcm⁵s²U]CCAAG, respectively. Ratios of modified oligonucleotides from the anticodon stem-loop of (G) Gln$^{CAA}$ (H) Glu$^{GAA}$ (I) Gly$^{GGA}$ isoacceptors expressed as percentages of their sum total. The figures are labeled as mean ± SD. Each time point is represented as a cumulative of two biological and two technical replicates (Appendix Table S6E).

their correlation with codon usage patterns using multivariate statistics (Fig 6C–H). Here, we identified 100 genes with the highest TE (HTE) and lowest TE (LTE) from the upper left and lower right quadrants, respectively, of scatterplots of normalized mRNA and

protein fold-change data at 40 and 46 h time points (Fig 6C and D). In PLS regression analysis of codon usage patterns of HTE and LTE genes, the scores plot for 40 h shows a weak discrimination of HTE genes from LTE genes (Fig 6E), while there was a much stronger

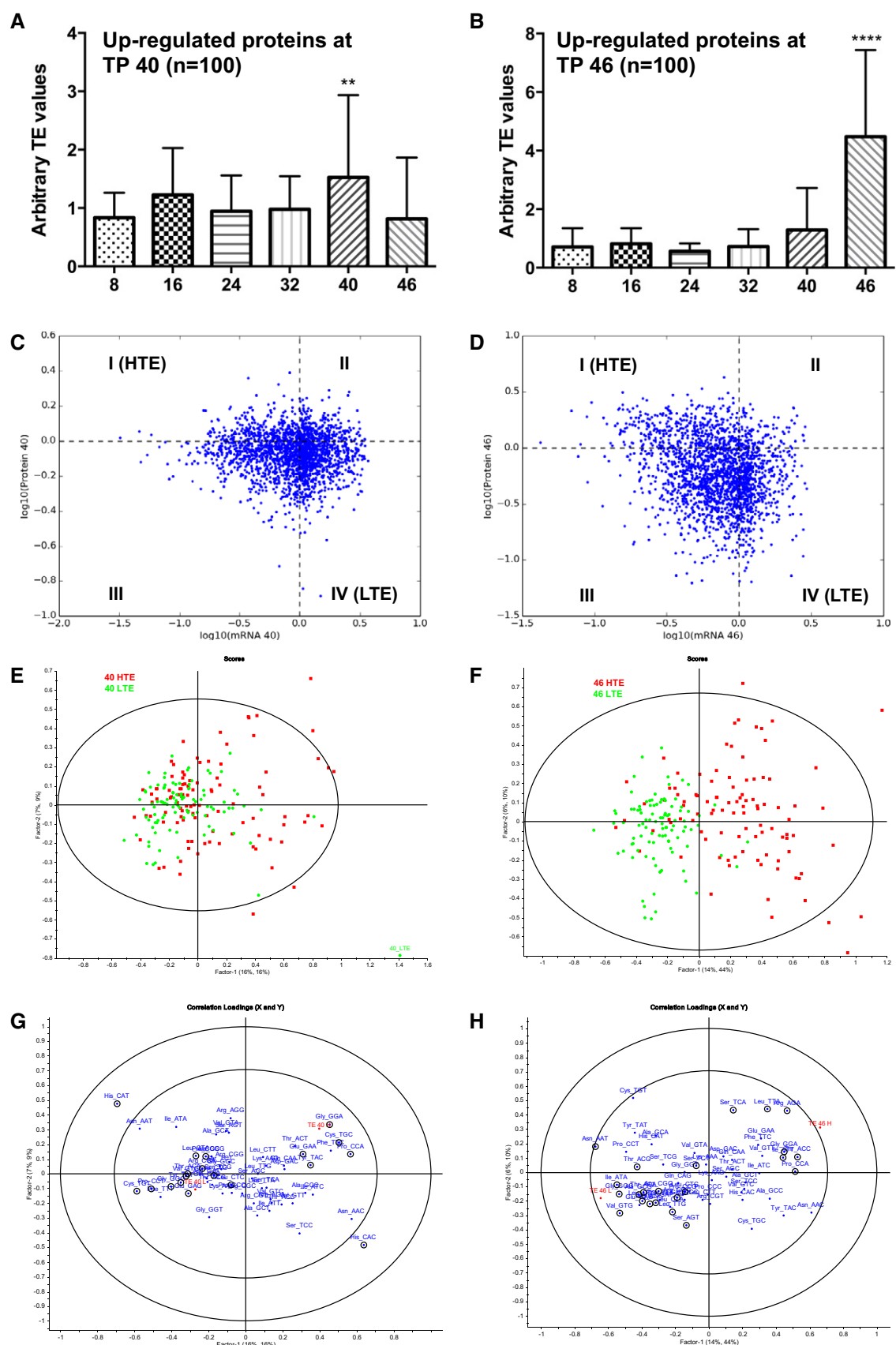

**Figure 6.**

**Figure 6. Highly up-regulated proteins in late IDC are not only codon-biased but also efficiently translated.**

A, B    Comparison of arbitrary translation efficiency (TE) values of (A) late trophozoite stage (TP 40) and (B) schizont stage (TP 46) proteins across all experimental time points. The arbitrary TE is defined here as the normalized protein abundance per mRNA abundance to mimic the translational output per mRNA copy in a biological system. The top 5% up-regulated proteins (n = 100) at (A) late trophozoite stage (TP 40) and (B) schizont stage (TP 46) were used for this analysis. Differences among all groups were first tested by one-way ANOVA analysis, followed by Dunnett's test as multiple comparison post-test to find significant differences between pairs of (A) TP 40 or (B) TP 46 and other time points. $P < 0.01$ is denoted as **. $P < 0.0001$ is denoted as ****. The figures are labeled as mean $\pm$ SD.

C–H    Scatterplots of mRNA and protein fold-changes (log-transformed) at time points (C) 40 h and (D) 46 h in the erythrocytic cycle. Quadrants are designated with Roman numerals. Data points falling into quadrant I are designated as high translationally efficient (HTE) proteins while those falling into quadrant IV are designated low translationally efficient (LTE) proteins. PLS regression analysis of codon usage of HTE (n = 100) versus LTE (n = 100) proteins at (E and G) TP 40 and (F and H) TP 46. (E and F) Visualized by scores. Eclipse in represents Hoteling $T^2$ limit at *P*-value of 0.01 (*F*-test). (G and H) Visualized by *x, y* correlation loadings. Outer and inner ellipses indicate 100 and 50% explained variance, respectively. Percentages of observed variances explained by the PC-1 and PC-2 are indicated in parentheses along their respective axes.

separation of HTE and LTE genes at 46 h (Fig 6F). The loadings plots for these PLS analyses (Fig 6G and H) reveal that most of the same pairs of synonymous codons associated with highly up- and down-regulated proteins at 40 and 46 h in the proteomics analysis (Fig 4C; Appendix Table S3) are also strongly associated with high TE (Fig 6G and H). For example, at 40 h, Gly$^{GGA}$, Cys$^{TGC}$, and Glu$^{GAA}$ are associated with HTE, while Gly$^{GGT}$, Cys$^{TGT}$, and Glu$^{GAG}$ are linked to LTE (Fig 6G). Similarly, at 46 h, Ser$^{TCA}$ and Val$^{GTT}$ are associated with HTE versus Ser$^{AGT}$ and Val$^{GTG}$ associated with LTE (Fig 6H). At both time points, Pro$^{CCA}$ and Pro$^{CCT}$ are linked to HTE and LTE, respectively. This comparison points to a strong association between codon usage patterns and translation efficiency at late stages of the *P. falciparum* IDC and strengthens the translational control model of tRNA modification reprogramming linked to an alternative genetic code for families of stage-critical genes.

# Discussion

Here, we present evidence for a new system of translational control of gene expression in malaria parasites. The model posits that, during the IDC, reprogramming of tRNA modifications coordinates with a system of synonymous codon usage to modulate the translational efficiency of stage-specific proteins. Recent studies point to the importance of specific instances of post-transcriptional regulation of gene expression in *P. falciparum* to explain the well-established delay between stage-specific transcription and translation in the IDC (Le Roch *et al*, 2004; Hall *et al*, 2005; Foth *et al*, 2008, 2011; Bunnik *et al*, 2013). Polysome profiling by Bunnik *et al* (2013) posits translational repression at the upstream open reading frames as a mechanism of control (Bunnik *et al*, 2013). Not mutually exclusive, our translational efficiency model in malaria parasites builds on observations in yeast and bacteria (Chan *et al*, 2010, 2012, 2015; Deng *et al*, 2015; Endres *et al*, 2015; Chionh *et al*, 2016), all of which point to coordinated behavior between the dozens of modified ribonucleosides in the tRNA epitranscriptome and an alternative genetic code of biased use of synonymous codons, which modulate translation efficiency to meet the needs of the cell. This "code of codons" is evident from the occurrence of pairs of synonymous codons differentially enriched in highly up- and down-regulated proteins at specific IDC stage (Fig 4), with no relationship to traditional notions of codon optimality or rarity (Appendix Tables S2 and S5; Sharp & Li, 1987). The codons enriched in highly up-regulated genes are significantly associated with wobble-modified tRNA isoacceptors that read the enriched codons, while tRNAs reading the

synonymous codon partners are enriched in down-regulated genes are not wobble-modified (Figs 4 and 5). This is more than coincidence, and studies in both yeast and bacteria point to direct cause-and-effect linkages between the modified tRNAs and the efficiency of translation of proteins from genes with matching codon biases (Chan *et al*, 2010, 2012, 2015; Dedon & Begley, 2014; Su *et al*, 2014; Deng *et al*, 2015; Endres *et al*, 2015; Chionh *et al*, 2016; Doyle *et al*, 2016).

## tRNA modification changes and codon-biased translation

A deeper analysis of the simplified tRNA system in *P. falciparum* supports this model for translational modulation of gene expression. Malaria parasites are unique in the AT richness of their genomes and the paucity of tRNA isoacceptors (45) to read > 60 codons, which necessitates "repurposing" of tRNA isoacceptors to differentiate synonymous codons. Here, our observations support the idea that "reprogramming" of wobble modifications achieves this codon selectivity. Ribonucleosides on the wobble position (34) of cytoplasmic tRNAs, particularly U-ending anticodons, are almost always modified to generate an adequate repertoire of anticodon structures and chemistries required for accurate decoding (Agris, 2004; Sprinzl & Vassilenko, 2005). This demand has resulted in a large repertoire of hyper-modified U ribonucleosides, such as mcm$^5$U, mcm$^5$s$^2$U, and ncm$^5$U, which can only be found at the wobble position in eukaryotic tRNAs and are highly conserved in their function to restrict tRNA binding to A-ending codons (Björk *et al*, 1999; Agris *et al*, 2007; Waas *et al*, 2007; Johansson *et al*, 2008; Phizicky & Hopper, 2010; Jackman & Alfonzo, 2013). Here, we have observed reprogramming of wobble U modifications to read A-ending codons strikingly over-used in up-regulated genes (Arg$^{AG\underline{A}}$, Gly$^{GG\underline{A}}$, Glu$^{GA\underline{A}}$, Pro$^{CC\underline{A}}$, Ser$^{TC\underline{A}}$). Although the modification of anticodon wobble A to I facilitates reading U, C, and A, the energetics of I:U and I:C pairs are more favorable than I:A (Agris *et al*, 2007).

A similar association between tRNA modification changes and codon-biased translation is apparent for Asn, His, Tyr, and Cys codons that are not read by wobble-modified tRNAs and may be controlled by changes in modifications at other positions. For each of these amino acids, there is a single tRNA isoacceptor that decodes two synonymous codons that we found to be differentially enriched in up- and down-regulated proteins at TP 40 and 46 (Appendix Table S3 shaded in gray). Relative to down-regulated proteins, we observed up-regulation of proteins from genes enriched with the C-ending codons that utilize Watson–Crick base pairing (C: G) for codon–anticodon interactions (Asn$^{AA\underline{C}}$, His$^{CA\underline{C}}$, Tyr$^{TA\underline{C}}$, and

**Figure 7.  Potential roles of tRNA modifications in asexual blood stage development.**

tRNA modifications orchestrate parasite development at a translational level in a collective model whereby: (i) High levels of Am and s²U at ring stage stabilize the "hypomodified" tRNA molecules or serve as a precursor for the subsequent modifications; (ii) a synchronous increase in 22 of the 28 modifications serves to signify tRNA maturation. This parallels up-regulation of translational and metabolic activities through the ring-to-trophozoite transition depicted by the graphs showing peaks of synthesis of DNA, RNA, and protein (adapted from de Rojas & Wasserman, 1985); (iii) increase in the levels of particular wobble modifications enhances the recognition to specific codon(s) out of other synonymous codons. This results in selective translation of codon-biased transcripts which manifests in the codon usages of the late trophozoite and schizont proteomes.

Cys$^{TG\underline{C}}$). Their anti-correlated counterparts in down-regulated genes are T-ending codons that are decoded by wobble U:G interactions (Asn$^{AA\underline{T}}$, His$^{CA\underline{T}}$, Tyr$^{TA\underline{T}}$, and Cys$^{TG\underline{T}}$). A key feature of these tRNAs is that they are not known to be modified at the wobble position, at least in yeast. In *P. falciparum,* there is only one tRNA isoacceptor available to decode synonymous codon sets Asn$^{AA\underline{T}}$/Asn$^{AA\underline{C}}$, His$^{CA\underline{T}}$/His$^{CA\underline{C}}$, Tyr$^{TA\underline{T}}$/Tyr$^{TA\underline{C}}$, and Cys$^{TG\underline{T}}$/Cys$^{TG\underline{C}}$ (Appendix Table S5). We found that highly expressed genes for late-stage proteins have biased use of C-ending codons Asn$^{AA\underline{C}}$, His$^{CA\underline{C}}$, Tyr$^{TA\underline{C}}$, and Cys$^{TG\underline{C}}$ (Fig 4C). These codons utilize canonical (G34:C3) base pairing to their respective cognate anticodon, instead of the T-ending codons Asn$^{AA\underline{T}}$, His$^{CA\underline{T}}$, Tyr$^{TA\underline{T}}$, and Cys$^{TG\underline{T}}$ that use wobble anti-codon:codon (G34:U3) interactions (Appendix Table S5).

Since the wobble G of these decoding tRNA species (tRNA$^{Asn(\underline{G}UU)}$, tRNA$^{His(\underline{G}UG)}$, tRNA$^{Tyr(\underline{G}ΨA)}$, tRNA$^{Cys(\underline{G}CA)}$) is not known to be modified, their codon–anticodon interactions may be regulated by non-wobble modifications, a model supported by our observations of IDC stage-specific tRNA modification reprogramming. Here, we point to the fact that modifications at position 37 also greatly affect codon–anticodon interactions. For example, t⁶A is required at position 37 of tRNAs reading ANN codons. In support of this idea, we observed that position 37 modifications t⁶A and ms²t⁶A are highly up-regulated during the second half of the IDC (Fig 2A). Though highly unlikely, it is possible that these tRNAs have yet-to-be detected wobble modifications in *P. falciparum*. Here, we speculate that canonical G34:C3 base pairing confers greater elongation

efficiency than the wobble G34:U3 during protein translation, though the G34:C3 and G34:U3 interactions have been argued to be isomorphic (Agris *et al*, 2007).

**tRNA maturation and parasite development**

In addition to tRNA modification changes associated with codon-biased translation, our analysis also revealed another facet of stage-specific tRNA modification reprogramming: simultaneous up-regulation of the majority of modified ribonucleosides in the ring-to-trophozoite transition. These changes are consistent with bulk maturation of tRNA in preparation for the ramp-up of translational machinery at the trophozoite–schizont border. The parasites develop from slow-growing, metabolically less-active ring forms to fast-growing metabolically active trophozoites and steadily grow in size as the schizonts mature (Gruring *et al*, 2011). The rapidly growing trophozoite stage is associated with significant metabolic activity including glycolysis, hemoglobin digestion, and macromolecular (DNA and protein) biosynthesis (de Rojas & Wasserman, 1985; Olszewski *et al*, 2009; Gruring *et al*, 2011; Lakshmanan *et al*, 2011; Lamour *et al*, 2014). While there have been no measurements of translational efficiency across the IDC, protein synthesis and translational activities in the second half of the developmental cycle are presumably higher than in the ring stage. Given the important roles of tRNA modifications in maintaining translational accuracy and efficiency, the observed up-regulation of tRNA modifications starting at the trophozoite stage is consistent with higher

translational activities during a fast-growing period. This suggests that parasites have distinct phases of tRNA modification levels at different stages to meet the "just-in-time" biological and translational needs.

### tRNA modifications associated with the IDC ring stage

Am and $s^2U$ modifications behave counter-cyclically to the bulk increase in tRNA modifications in the ring-to-trophozoite transition, with a gradual decrease across the IDC cycle (Fig 2A). Am occurs in the stem-loop of several eukaryotic tRNAs (Agris *et al*, 2007) and stabilizes tRNA structure (Kawai *et al*, 1992; Jackman & Alfonzo, 2013), with Am linked to the stability of tRNAs in hyperthermophiles (McCloskey *et al*, 2001). Although the distribution and location of Am in *P. falciparum* tRNA remains unknown, it is possible that the higher level of Am during the early stage of IDC functions to maintain the stability of "hypomodified" tRNAs in ring stage. 2-Thiolation of uridine typically occurs at the wobble position as a precursor or intermediate in the formation $mcm^5s^2U$ in $tRNA^{Gln}$ $^{(UUG)}$, $tRNA^{Glu(UUC)}$, and $tRNA^{Lys(UUU)}$ in all three domains of life (Klassen *et al*, 2015). The order of formation of $s^2U$ and $mcm^5$ in the synthesis of $mcm^5s^2U$ appears to vary among organisms (Nakai *et al*, 2008; Noma *et al*, 2009; Han *et al*, 2015), so it is uncertain if the abundance of $s^2U$ in the ring stage serves as a reservoir precursor to $mcm^5s^2U$ as the level of the latter modification increases throughout the IDC. While the biochemistry of $mcm^5s^2U$ formation in *P. falciparum* remains elusive, the results of our studies point to an epitranscriptomic model for protein synthesis in the parasite.

### An integrated model for tRNA epitranscriptomics in the *P. falciparum* IDC

Integrating the epitranscriptomic, proteomic, and genomic analyses presented here leads to a testable model for translational modulation of gene expression in the malaria IDC. As depicted in Fig 7, the system of ~30 tRNA modifications undergoes three stage-specific signature "reprogramming" events tied to regulation of the parasite IDC (i): Up-regulation of Am and $s^2U$ in the ring stage heralds subsequent tRNA maturation in later IDC stages; (ii) a "rising tide" of modification levels in the ring-to-trophozoite transition period signifies tRNA maturation that parallels up-regulation of translational and metabolic activities as the parasite matures; and (iii) the reprogramming of specific wobble tRNA modifications ($mcm^5U$, $mcm^5s^2U$, $ncm^5U$, Gm, and I) at the late stage of parasite development coordinates with an alternative genetic code of biased use of synonymous codons to up- and down-regulate expression of critical and unneeded proteins, respectively. These results shed light on the complexity of tRNA epitranscriptomics and the transcendent links between the epitranscriptome, proteome, and genome in the developmental cycle of malaria parasites.

## Materials and Methods

### Reagents

Unless otherwise stated, chemical reagents were purchased from Sigma-Aldrich, Purelink RNA extraction kits were purchased from Life Technologies, LC-MS grade solvents from Thermo Fisher, and RNA and DNA oligonucleotides from Integrated DNA Technologies.

### Synchronization and sampling of *P. falciparum* cultures *in vitro*

Cultures of *P. falciparum* 3D7 (Gardner *et al*, 2002) in asexual stages were grown in human red blood cells (RBCs) in RPMI 1640 medium (Invitrogen) supplemented with 0.25% Albumax II (Gibco, Life Technologies), 2 g/l sodium bicarbonate (Sigma), 0.1 mM hypoxanthine (Sigma), and 50 mg/l gentamycin (Gibco, Life Technologies). The cultures were incubated at 37°C, with 5% $CO_2$ and 3% $O_2$. The cultures were tested for mycoplasma contamination using the MycoAlertTM Mycoplasma Detection Kit (Lonza) prior to starting an experiment, due to potential artifacts caused not only by microbial influences on parasite biology but also contributions of bacterial RNA, and associated modifications, to the parasite RNA. Microscopic inspections were routinely performed to check for culture contamination. Cells were synchronized by repeated treatment with 5% L-sorbitol (Sigma-Aldrich) for two to three generations, to enrich for ring-form infected and uninfected RBCs. After each sorbitol treatment, the mixture was centrifuged at $600 \times g$ (acceleration, 9; deceleration, 2) for 3 min to remove sorbitol and lysed infected RBCs in the supernatant. The pelleted RBCs were washed twice with incomplete RPMI 1640. Following this enrichment process, highly synchronized cells were used to re-infect human RBCs, with the time point (TP) 0 defined as the starting point of the invasion. Immediately after the infection event (TP 0), parasitized RBCs were collected at 8, 16, 24, 32, 40, and 46 h post-infection (hpi) over the course of IDC (defined as TP 8, TP 16, TP 24, TP 32, TP 40, and TP 46, respectively). At each time point, thin smears were analyzed by microscopy to ensure that culture synchrony was above 80%. To minimize host RNA contamination, parasites were purified by lysing human RBCs with 0.15% saponin (Sigma) to selectively rupture RBC membranes (Gruenberg & Sherman, 1983) and release the parasite, and the released parasites washed three times with ice-cold PBS. Purified parasites were homogenized with 5 volumes of TRIzol Reagent (Invitrogen), followed by a 5-min incubation at ambient temperature before flash freezing in liquid nitrogen.

### RNA extraction from homogenized cell lysates

The quality and purity of an RNA analyte are essential for characterizing its complement of modified ribonucleosides. As the first feature of the RNA modification analysis platform, we established a rigorous protocol for isolation, purification, and quality assessments of *P. falciparum* tRNA prior to downstream LC-MS/MS analysis (Appendix Fig S1A). Chloroform (Sigma) was added to the TRIzol-homogenized cell lysates at one-fifth lysate volume, and the mixture was incubated at ambient temperature for 3 min. The mixture was centrifuged at $12,000 \times g$ for 15 min at 4°C, and the aqueous phase was collected. Next, the aqueous phase was adjusted to 35% v/v ethanol, followed by RNA extraction using the PureLink™ miRNA Isolation Kit (Invitrogen) according to the manufacturer's instructions. A sequential isolation protocol was adopted to enrich the yield of small RNA species, particularly tRNA using 70% v/v ethanol (Merck) for further experiments. The quantity and quality of RNA were assessed using a Bioanalyzer with RNA 6000 Small RNA chips, Pico chips, and Nano chips (Agilent Technologies).

To ensure the effectiveness of saponin treatment in minimizing RNA derived from the host RBCs, we compared the RNA profiles of parasite-free RBC controls with and without saponin lysis. The RNA fraction isolated from RBCs not treated with saponin was found to contain host RNA—mostly small RNA species (tRNA and 5S rRNA), along with minor amounts of human 18S rRNA (~1,900 nt) and 28S rRNA (~5,000 nt; Appendix Fig S1B, blue). Importantly, no detectable RNA was present in saponin-lysed RBC control (Appendix Fig S1B, red), confirming the effectiveness of the protocol in diminishing host cell RNA contamination. The quality of the resulting RNA extracted from the parasite is shown in Appendix Fig S1B (black), with sharp peaks for the tRNA (65–85 nt), 5S rRNA (~120 nt), 5.8S rRNA (~150 nt), 18S rRNA (~2,000 nt), and 28S rRNA (~4,000 nt) expected for *P. falciparum*. Essentially, no trace of human 18S rRNA and 28S rRNA was found in the parasite RNA fraction, further excluding the possibility of host RNA contamination. We also observed smaller peaks (0.5–2% of the integrated peak area of 18S and 28S rRNAs) corresponding to the sizes of *P. falciparum* apicoplast 16S rRNA (~1,500 nt) and 23S rRNA (~3,000 nt; Appendix Fig S1B, black). These observations are consistent with previous reports suggesting that *P. falciparum* has a complete set of eukaryotic RNA species with no evidence of miRNA, while the organelle apicoplast contains rRNA of plastid and bacterial origin (Waters *et al*, 1989, 1995; Gardner *et al*, 1993, 2002; Li *et al*, 1994; John Rogers *et al*, 1996; Hughes *et al*, 2010; Jackson *et al*, 2011).

Subsequently, we optimized the protocol to enrich the small RNA fraction (< 200 nt) from total RNA populations through selective binding of RNA molecules to silica-based membrane of spin cartridges. All of the isolated fractions were then checked for degradation and purity, as well as comparison to parasite-free RBC controls. RNA eluted from the first spin cartridge (35% ethanol, solid-phase extraction I) contains 28S and 18S rRNA with a ratio close to 2, along with 5.8S and 5S rRNA (Appendix Fig S1C). On the other hand, the RNA fraction retained on the membrane of the second spin cartridge (70% ethanol, solid-phase extraction II) consisted of highly purified tRNA with the occasional presence of small quantities of 5S rRNA (0.5–5% of the integrated tRNA peak area; Appendix Fig S1D). Although the purity of the isolated tRNA fraction is relatively high, the potential for artifacts caused by even minor amounts of small RNA fragments and 5S rRNA necessitated that we perform an additional HPLC purification step.

### RNA purification by size exclusion chromatography

The second critical step of our tRNA modification analysis platform involves isolation of the population of tRNA molecules to homogeneity to ensure the characterized modifications are solely from tRNA. Here, we used size exclusion (SE) HPLC to purify all different classes of non-coding RNA (ncRNA) based on sequence length and secondary structure, with emphasis on preserving RNA modifications during the purification (Chionh *et al*, 2013). To purify tRNA to homogeneity, one-dimensional size exclusion chromatography (SEC) was employed using a Bio SEC-3 300 Å (ID 7.8 mm, 300 mm) column (Agilent Technologies). Ammonium acetate (100 mM, pH 7.0, Sigma-Aldrich) was used as the mobile phase. Isocratic separations were performed at 1 ml/min for 22 min under partially denaturing conditions at 60°C on an Agilent 1200 HPLC system, with the RNA elution monitored by diode array absorbance spectroscopy

(200–300 nm) at 10-nm intervals. The tRNA-containing peak was collected (Agilent 1260 Infinity Fraction Collector) and desalted and concentrated with a Vivacon 500 2K MWCO spin column (Sartorious Stedium) according to the manufacturer's instructions. The quality and quantity of tRNA were assessed using the Agilent Bioanalyzer RNA 6000 Small RNA chip. As shown in Appendix Fig S1E, there were no detectable small RNA species other than tRNA, which demonstrates its high purity (95–99%) and integrity after HPLC purification. Concentrations of RNA samples were determined by signal intensity of RiboGreen fluorescent dye bound to the RNA molecules (Agilent Bioanalyzer) and UV absorbance at 260 nm (NanoDrop 1000 Spectrometer, Thermo Scientific).

### LC-MS/MS identification of modified ribonucleosides in tRNA

The characterization of ribonucleosides involved a combination of top-down and bottom-up approaches, including neutral loss scan (NLS), molecular feature extraction (MFE), multiple reaction monitoring (MRM), and target ion extraction (TIE) analyses to define 28 distinct chemical modifications in purified tRNA (Fig 1A, Appendix Table S1). In all cases, purified *P. falciparum* tRNA (3 µg) was hydrolyzed enzymatically as described (Su *et al*, 2014). $[^{15}N]_5$-2-deoxyadenosine ($[^{15}N]$-dA) was added to the digested tRNA samples as the internal standard to account for variations in sample handling and MS response. For each LC-MS/MS experiment, 0.2 µg of digested tRNA was injected. Hypersil GOLD aQ column (100 × 2.1 mm, 1.9 µm, Thermo Scientific) was used to resolve the digested RNA in a two-buffer eluent system with buffer A consisting of ultrapure water with 0.1% (v/v) formic acid and buffer B consisting of acetonitrile with 0.1% (v/v) formic acid. All solvents were LC-MS grade. HPLC was performed on Agilent 1290 uHPLC system at 25°C with a flow rate of 0.3 ml/min. The gradient of acetonitrile with 0.1% (v/v) formic acid was as follows: 0–12 min, held at 0%; 12–15.3 min, 0–1%; 15.3–18.7 min, 1–6%; 18.7–20 min, held at 6%; 20–24 min, 6–100%; 24–27.3 min, held at 100%; 27.3–28 min, 100–0%; 28–41 min, 0%. The same HPLC system was coupled to an Agilent 6460 QQQ or an Agilent 6520 quadrupole time-of-flight (QTOF) mass spectrometer.

As a first approach for tRNA modification discovery, semi-targeted NLS using the LC-QQQ system was applied to search for potential modified ribonucleosides in digested tRNA hydrolysate, with a parallel examination of a mock digest sample prepared from parasite-free RBC to account for artifacts. These analyses were performed on the LC-QQQ system with ESI Jetstream ionization operated in positive ion mode. The voltages and source gas parameters were as follows: gas temperature, 350°C; gas flow, 5 l/min; nebulizer, 40 psi; sheath gas temperature, 325°C; sheath gas flow, 7 l/min; capillary voltage, 4,000 V and nozzle voltage, 500 V.

In NLS analysis, a characteristic cleavage of the glycosidic bond between a nucleobase and either ribose or 2′-O-methylribose during CID causes the loss of uncharged (hence, neutral) fragment of either 132 or 146 atomic mass units (amu), respectively. By searching the spectra for neutral loss of 132 or 146 amu in the *m/z* range of 200–700, NLS analysis selected all molecules containing ribose or a 2′-O-methylribose—the defining molecular signatures of ribonucleosides, except for pseudouridine (Ψ), whose presence was verified by CID fragmentation of *m/z* 125 or 209 (Dudley *et al*, 2005). The ions detected in NLS were selected for validation by multiple reaction

monitoring (MRM) using the same HPLC and MS parameters. The exact molecular weights of molecular ions and CID fragments were determined by LC-QTOF with ESI ionization. The LC-QTOF MS system operated in positive ion mode and scanned for ions from $m/z$ 100 to 1,700 with the following parameters: gas temperature, 325°C; drying gas, 5 l/min; nebulizer, 30 psi; and capillary voltage, 3,500 V.

At the same time, the MFE and TIE approaches used in the LC-QTOF MS analysis provided the exact mass of the ribose- or 2′-O-methylribose-containing precursor ions. The processes of untargeted molecular feature finding and targeted ion extraction were performed using the Agilent MassHunter Workstation vB05.00, which allowed us to compare the accurate masses of observed precursor ions to the theoretical molecular masses of those known modifications cataloged in the chemical databases (Cantara *et al*, 2011; Machnicka *et al*, 2013). The identities of many of the individual ribonucleosides were established using commercially available standards (Appendix Table S1). For the putative modified ribonucleosides for which standards were not commercially available, we compared their observed $m/z$ values with the theoretical molecular masses reported in Modomics (http://modomics.genesilico.pl/) and the RNA modifications database (http://mods.rna.albany.edu/), in parallel with CID fragmentation analysis (Appendix Table S1). All the annotations were checked for the exclusion of isotopic compounds and salt adducts. Only the compounds reproducibly detected in all biological replicates were validated and are listed in Appendix Table S1.

### LC-MS/MS quantification of tRNA modifications across the IDC

Purified *P. falciparum* tRNA (3 μg) from three biological replicates of selected time points was hydrolyzed enzymatically as described in Su *et al* (2014). [$^{15}$N]$_5$-deoxyadenosine ([$^{15}$N]-dA) was added to the digested tRNA samples as an internal standard to account for sampling and machine variations. Hypersil GOLD aQ column (100 × 2.1 mm, 1.9 μm, Thermo Scientific) was used to resolve the resulting ribonucleosides in a two-buffer eluent system, with buffer A consisting of water with 0.1% (v/v) formic acid and buffer B consisting of acetonitrile with 0.1% (v/v) formic acid. All solvents used were LC-MS grade. HPLC was performed at 25°C with a flow rate of 0.3 ml/min. The gradient of acetonitrile with 0.1% (v/v) formic acid was as follows: 0–12 min, held at 0%; 12–15.3 min, 0–1%; 15.3–18.7 min, 1–6%; 18.7–20 min, held at 6%; 20–24 min, 6–100%; 24–27.3 min, held at 100%; 27.3–28 min, 100–0%; 28–41 min, 0%. The HPLC column was directly connected to an Agilent 6460 triple quadrupole mass spectrometer (LC-MS/MS) with ESI Jetstream ionization operated in positive ion mode. The voltages and source gas parameters were as follows: gas temperature, 350°C; gas flow, 5 l/min; nebulizer, 40 psi; sheath gas temperature, 325°C; sheath gas flow, 7 l/min; capillary voltage, 4,000 V and nozzle voltage, 500 V. The pre-determined molecular transition ions listed in Appendix Table S1 were quantified in multiple reaction monitoring (MRM) mode. The ions were monitored at pre-defined times based on LC retention time between 0.5 and 28 min. The dwell time for each ribonucleoside was 100 ms. The samples for three independent biological replicates were analyzed in the same LC-MS/MS analysis to minimize batch-to-batch fluctuation in MS sensitivity. Quantitative comparisons between biological replicates from various time

points were made possible by correcting for biological variation in total tRNA quantities by dividing the raw peak area of the ribonucleoside to the total area under curve (AUC) of the four canonical ribonucleosides (C, U, A, and G). For the mapping and quantitative studies, purified tRNA (500 μg) from ring, trophozoite, and schizonts was digested using 2U RNase T1 (Thermo) in RNA structure buffer for 2 h at 37°C followed by 60°C for 4 h. The enzymes were cleaned up using a 10K spin filter (Omega Nanosep), and the sample was desalted and concentrated using C18 Ziptip solid-phase extraction cartridges (Sigma).

### Protein extraction and processing

Biological triplicates of *P. falciparum* cells at six experimental time points (TP 8, 16, 24, 32, 40, and 46) across the IDC were isolated as follows. Parasite-infected RBCs were centrifuged at 600 × *g* for 3 min at ambient temperature to collect the erythrocyte pellet. To minimize host cell RNA contamination, parasites were purified by lysing human RBCs with 0.15% (w/v) saponin (Sigma) solution containing ethylenediaminetetraacetic acid (EDTA)-free protease inhibitor (Roche Applied Science). The lysate was resuspended repeatedly on ice for 4 min to ensure complete lysis of RBC membrane. After three washes with ice-cold PBS, purified parasites were homogenized with 5 volumes of TRIzol Reagent (Invitrogen), followed by 5-min incubation at ambient temperature before flash freezing in liquid nitrogen or proceeding to parasite lysis step. The parasite pellet was resuspended in buffer A consisting of 50 mM triethylammonium bicarbonate (TEAB) (Fluka), 6 M urea (Sigma), and a mixture of EDTA-free protease inhibitor cocktail (Roche Applied Science). The suspension was homogenized by sonication (Vibracell, USA) for 10 min on ice at 25% amplitude with pulses of 2 s on, 3 s off, resulting in 4 min total pulse-on time. The insoluble material was then centrifuged at 16,000 *g* for 30 min at 4°C. The soluble fraction of the cell lysate was mixed with 4 volumes of cold acetone (Fluka) and incubated at −20°C overnight for protein precipitation. The tubes were then centrifuged for 10 min at 15,000 × *g* and 4°C, and the supernatant was discarded. The protein pellets were washed once with cold acetone followed by centrifugation for 10 min at 15,000 × *g* at 4°C. After discarding the supernatant, the protein pellets were allowed to air-dry. The dried protein pellets were dissolved with 1% sodium dodecyl sulfate (SDS) (Sigma) in 50 mM triethylammonium bicarbonate (TEAB) buffer (pH 8.5) (Sigma). The Bradford (Pierce BioScience) assay was used to determine protein concentrations, and SDS-polyacrylamide gel electrophoresis (12% polyacrylamide gels) was used to examine protein quality and profile.

### In-gel protein fractionation, reduction, alkylation, and tryptic digestion

Protein from each sample (100 μg) was mixed with 4× loading buffer (Bio-Rad) and heated at 95°C for 5 min before loading on a gel. Protein separation was performed on a precast Mini Protean 4–20% TGX gel (Bio-Rad) resolved at 80 V for 90 min. The gel was subjected to Coomassie blue (Bio-Rad) staining and destaining to visualize the protein bands. Each gel lane was cut into five equal fractions. Each fraction was sliced into 1 mm³ pieces and washed three times by shaking at ambient temperature for 10 min with 50%

acetonitrile in 25 mM TEAB buffer for complete destaining. The destained gel pieces were subjected to complete dehydration with 100% acetonitrile and briefly dried (1–2 min) with a vacuum concentrator. For reduction reaction, 10 mM of tris-(2-carboxyethyl) phosphine [TCEP] (Sigma) in 25 mM TEAB buffer was added to cover the dried gel pieces and incubated at 37°C for 1 h. The gel pieces were then dehydrated by adding 500 μl of acetonitrile, and the liquid was removed. For the alkylation reaction, 20 mM of iodoacetamide (IAA) (Sigma) with 25 mM TEAB was added to cover the gel pieces and incubated at ambient temperature in the dark for 45 min, subsequently, with 50% acetonitrile in 25 mM TEAB buffer to remove excess TCEP and IAA. Afterward, 100% acetonitrile was used to further dehydrate the gel pieces, followed by complete drying by vacuum centrifugation. For tryptic digestion, proteomics-grade trypsin (Sigma) at 10 ng/μl in 10% acetonitrile in 25 mM TEAB buffer was added to the dried gels to an approximate ratio of 1:30 (w/w) of trypsin to protein substrates. The tubes were incubated at 4°C for 30 min to enable adequate absorption of trypsin solution into the gel pieces. A sufficient amount of 10% acetonitrile in 25 mM TEAB buffer was then added to cover the rehydrated gel pieces, followed by an incubation for 16 h at 37°C to ensure complete digestion. The peptide-containing digest solution was collected, followed by re-extraction three times with 5% formic acid in 50% acetonitrile by vortexing for 15 min at ambient temperature, and dehydration of gel pieces with 100% acetonitrile. All the extracts were pooled into the initial digest solution. After that, the peptides were dried under vacuum and reconstituted with 30 μl of 100 mM TEAB buffer.

### iTRAQ labeling, solid-phase extraction, and peptide fractionation

Peptide digests were labeled with multiplex iTRAQ reagents according to the manufacturer's instructions (Applied Biosystems, USA). The peptide samples were labeled as follows: iTRAQ-113 – TP 8; iTRAQ-114 – TP 16; iTRAQ-115 – TP 24; iTRAQ-116 – TP 32; iTRAQ-118 – TP 40; iTRAQ-119 – TP 46. After iTRAQ labeling, the samples were combined and dried by vacuum centrifugation. The labeled peptides were dissolved with 0.1% trifluoroacetic acid (TFA) (Sigma) in 1% acetonitrile. For desalting using solid-phase extraction (SPE) procedure, Sep-Pak C18 cartridges (Waters) were attached to vacuum manifold and first equilibrated with 3 ml of 0.1% TFA in 70% acetonitrile and then flushed with 10 ml of 0.1% TFA in 1% acetonitrile. Next, peptide digest was added to the column and allowed binding to the C18 matrix, followed by washing with 10 ml of 0.1% TFA in 1% acetonitrile. After that, three times of elution were made with 0.1% TFA in 70% acetonitrile. The eluents were pooled and dried by vacuum centrifugation. The desalted peptides were reconstituted in immobilized pH gradient (IPG) buffer (GE Healthcare) without glycerol. For each sample that was pre-fractionated by in-gel separation, isoelectric focusing-based fractionation was performed from pH 3 to 10 over 12 wells on an Agilent 3100 OFFGEL fractionator according to the manufacturer's protocol (OG12PE00). All the 60 fractions (five in-gel fractions × 12 off-gel fractions) were collected and analyzed by nano-LC-MS/MS.

### LC-MS/MS for mapping tRNA modifications

Amide-HILIC chromatographic separation was performed on a TSK-gel Amide-80 column (2.0 mm ID × 150 mm, 3 μm particle size)

using a binary solvent system consisting of 8 mM ammonium acetate in ultrapure water (solvent A) and acetonitrile (solvent B). The flow rate was set to 0.1 ml/min. The gradient of solvent A was as follows: 0–2 min, held at 10% (v/v); 2–3 min, 10–15%; 3–5.5 min, 15–30%; 5.5–20.5 min, 30–60%; 20.5–25 min, 60–70%; 25–28 min, 70–10%. The HPLC column was maintained at 50°C. The parameters for the QToF mass spectrometer were as follows: gas temperature 325°C, gas flow 8 l/min, nebulizer 30 psi, $V_{cap}$ 3,800 V, Fragmentor 250 V, Skimmer 165 V, octopole RF peak of 750. Targeted MS2 performed every 5 V from collision energies from 25 to 45 V and products scanned from 100 to 1,500 $m/z$.

### LC-MS/MS analysis of the parasite proteome

iTRAQ proteomics experiments were performed on an Agilent 1200 nano-LC-Chip/MS interfaced to an Agilent 6510 QTOF LC/MS. The LC system consisted of a capillary pump for sample loading, a nano-flow pump, and a thermostated microwell-plate autosampler. The HPLC-Chip configuration consisted of a 160 nl enrichment column and a 150 mm × 75 μm analytical column (G4240-62001 Zorbax 300SB-C18). Mobile phases employed were as follows: 0.1% formic acid in water (solvent A) and 0.1% formic acid in acetonitrile (solvent B). A 120-min gradient LC separation was used with 10 min for column wash and equilibration between runs. Samples were loaded onto the enrichment column at 1% (v/v) B at flow rates of 3 μl/min. On the nanoflow pump, the gradient of solvent B was as follows: 0–1 min, held at 1% (v/v), flow rate from 0.4–0.2 μl/min; 1–101 min, 1–45%, flow rate held at 0.2 μl/min; 101–121 min, 45–75%, flow rate held at 0.2 μl/min; 121–122 min, 75–98%, flow rate from 0.2–0.4 μl/min; 122–126 min, held at 98%, flow rate held at 0.4 μl/min; 126–127 min, 98–1%, flow rate held at 0.4 μl/min; 127–130 min, held at 1%, flow rate held at 0.4 μl/min. LC-QTOF was operated at high resolution (4 GHz) in positive ion mode with the following source conditions: gas temperature 325°C, drying gas 5 l/min, fragmentor 225 V. Capillary voltage was adjusted between 1,500 and 2,100 V manually to achieve a steady spray. Data were acquired from 200 to 1,700 $m/z$ with an acquisition rate of five spectra/s in MS mode and from 50 to 2,200 $m/z$ with an acquisition rate of two spectra/s in MS/MS mode. LC/MS data were extracted and evaluated using the molecular feature extractor (MFE) algorithm in MassHunter Qualitative Analysis software (vB04.00). Test injections (2–3) from each fraction of the first technical replicate were made to optimize injection volumes for the second and third biological replicates for maximal extracted molecules with peptide-like features. For each fraction, the MFE list of molecular ions was exported and used to exclude the acquisition of spectra from these ions in subsequent runs. As such, every fraction from each biological replicate was run twice, first without and later with the exclusion list.

### Database analysis

Tandem mass spectra were extracted and deisotoped and the charge state deconvoluted using Spectrum Mill (Agilent; v B.04.00.127), and the spectra submitted using Spectrum Mill and X!Tandem (The GPM, thegpm.org; version CYCLONE (2010.12.01.1)) to search the UniProtKB/SWISS-PROT protein sequence databases for *P. falciparum* 3D7 (reference proteome: *Plasmodium falciparum* (isolate 3D7), 5353 entries, no isoforms) and *Home sapiens* (reference

proteome: *Home sapiens*, 68,511 entries, no isoforms) with tryptic digest fragments with an ion mass tolerance of 50 PPM and a parent ion tolerance of 20 PPM. Carbamidomethylation of cysteine and iTRAQ 8-plex of lysine, tyrosine, and the N-terminus were specified in Spectrum Mill and X!Tandem as fixed modifications. Ammonia loss of the N-terminus, deamidated of asparagine, and oxidation of methionine were specified in Spectrum Mill as variable modifications. Glu->pyro-Glu of the N-terminus ($-18.01$), ammonia loss of the N-terminus ($-17.03$), Gln->pyro-Glu of the N-terminus ($-17.03$), deamidated of asparagine and glutamine ($+0.98$), oxidation of methionine ($+15.99$), acetyl of lysine and the N-terminus ($+42.01$), carbamyl of lysine ($+43.01$), carbamidomethyl of cysteine ($+57.02$), carboxymethyl of cysteine ($+58.01$), phosphorylation of serine, threonine, and tyrosine ($+79.97$), and iTRAQ 8-plex of lysine, tyrosine, and the N-terminus ($+304.21$) were specified in X!Tandem as variable modifications.

### Criteria for protein identification

Scaffold (version Scaffold_4.4.5, Proteome Software Inc.) was used to validate MS/MS-based peptide and protein identifications. Peptide identifications were accepted if they could be established at < 1% false discovery rate (FDR) by the Scaffold Local FDR algorithm. Protein identifications were accepted if they could be established at > 99% probability and contained at least two identified peptides. Protein probabilities were assigned by the Protein Prophet algorithm (Nesvizhskii *et al*, 2003). Proteins that contained similar peptides and could not be differentiated based on MS/MS analysis alone were grouped to satisfy the principles of parsimony. Proteins sharing significant peptide evidence were grouped into clusters. The processed scaffold dataset used in the study is accessible as Appendix Table S7.

### Relative protein quantification by iTRAQ

Scaffold Q+ (version Scaffold_4.4.5, Proteome Software Inc.) was used to analyze iTRAQ-labeled based peptide quantitation and protein identifications. Peptide identifications were accepted if they could be established at < 1% FDR by the Scaffold Local FDR algorithm. Protein identifications were accepted if they could be established at > 99% probability and contained at least two identified peptides. Protein probabilities were assigned by the Protein Prophet algorithm (Nesvizhskii *et al*, 2003). Proteins sharing significant peptide evidence were grouped into clusters. Channels were corrected by the matrix in all samples according to the i-Tracker algorithm (Shadforth *et al*, 2005). Acquired intensities in the experiment were globally normalized across all acquisition runs. Individual quantitative samples were normalized within each acquisition run. Intensities for each peptide identified were normalized within the assigned protein. The reference channels were normalized to produce a 1:1 fold-change. Only those with reporter ion peak heights > 1% of the highest peak in a spectrum were for quantitation.

### Data processing and statistical analysis

Abundances of tRNA modifications were transformed to $\log_2$ ratios of modification levels in each time point relative to an arbitrary average control across all samples. Hierarchical clustering analysis (HCA) was performed with average linkage algorithm using centered Pearson correlation as similarity metric by Cluster 3.0. Microsoft Excel was used to generate heat map for visualization. Comparisons between two samples were made using the appropriate two-tailed Student's *t*-tests. Comparisons between multiple samples were determined using one-way ANOVA when comparing 1 factor. These statistical tests were performed using Prism 5 (GraphPad) or OriginPro 8 (OriginLab). Unless otherwise stated, all data are represented as arithmetic means ± SEM. To aid interpretation on statistical significances, $P < 0.05$ and $P < 0.01$ are denoted as * and **, respectively, unless otherwise stated. Levels of tRNA modifications for each time point were analyzed by principal component analysis (PCA) using non-linear iterative partial least squares (NIPALS) algorithm. Outliers which could cause over-fitting, if applicable, were removed by inspection of residual sample variances and leverages, as well as Hotelling's $T^2$ statistics. Marten's uncertainty test with optimal number of PCs was used to cross-validate the PCA model. Eigenvector-based multivariate statistics performed using UnscramblerX® (v10.3, Camo).

The pathway enrichment analysis of up-regulated proteins executed based on Malaria Parasite Metabolic Pathways (MPMP) and Gene Ontology (GO) annotated for *P. falciparum* (PlasmoDB v8.2) using in-house program developed by Prof Zbynek Bozdech Laboratory, School of Biological Sciences, NTU.

For descriptive interpretations of the relationships between codon usage (codon frequency) and up-regulated proteins at different time points, PCA was performed using NIPALS algorithm. The values of codon usage in synonymous codon choices of those proteins were retrieved from the pre-calculated genome-wide codon usage. Alternate start and stop codons were treated as categorical variables. Outliers which could cause over-fitting, if applicable, were removed by inspection of residual sample variances and leverages, as well as Hotelling's $T^2$ statistics. Marten's uncertainty test with optimal number of PCs was used to cross-validate the PCA model. Eigenvector-based multivariate statistics performed using UnscramblerX® (v10.3, Camo). Interpretations of the relationships between codon usage predictors (codon frequency) and protein up- or down-regulation ($\log_2$ median fold-change) were analyzed by partial least squares regression (PLS). Outliers which could cause over-fitting, if applicable, were removed by inspection of variable residuals and leverages, as well as Hotelling's $T^2$ statistics. Marten's uncertainty test with optimal number of PCs was used to cross-validate the PLS model. Eigenvector-based multivariate statistics performed using UnscramblerX® (v10.3, Camo).

### Translational efficiency calculations

An arbitrary index of translational efficiency (TE) was quantified as the ratio of the fold-change value for a protein to the fold-change value for the corresponding mRNA at a specific time point. This represents an estimate of translational output per mRNA copy and is similar to other published TE indices (Schwanhausser *et al*, 2011). The quantitative mRNA abundances data for these proteins were retrieved from a published time-course RNA-seq data at 8-h resolution across the IDC, which used the same *P. falciparum* strain (isolate 3D7), same culturing conditions, and synchronization

method as in our experiments (Otto *et al*, 2010; data downloaded from PlasmoDB, http://www.plasmodb.org). The sampling time points of the RNA-seq study are equivalent to our tRNA modifications and quantitative proteomics experiments, with a 2-h difference in the last time point, whereby cultures were sampled at 48 h instead of 46 h in our studies. Proteomics (here) and published RNA-seq (Otto *et al*, 2010) data were normalized to account for different scales.

## Data availability

The processed tRNA modification quantitation data are available in Appendix Table S6, and the processed proteomics data from scaffold are available in Appendix Table S7. The raw data are available in the following database:

• Proteomics data, tRNA modification quantitation, and mapping data: CHORUS 1443 (https://chorusproject.org/pages/dashboard.html#/projects/all/1443/experiments).

**Expanded View** for this article is available online.

## Acknowledgements
This work is supported by the National Research Foundation Singapore under its Singapore-MIT Alliance for Research and Technology (SMART) Centre, Infectious Disease IRG. C.S.N., A.S., and Y.H.C. acknowledge financial support from the Singapore-MIT Alliance (SMA) Graduate Fellowships.

## Author contributions
CSN, PCD, and PRP conceived the original idea and designed the study. CSN, AS, and QN designed and performed all the experiments and data analysis. CSN, YA, IRB, CG, and YHC were involved in the proteomics study. CSN, AS, PCD, and PRP wrote the paper, and all the authors checked through the manuscript and analyzed the data.

## Conflict of interest
The authors declare that they have no conflict of interest.

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
