## [Review Process File · Molecular Systems Biology]

tRNA epitranscriptomics and biased codon are linked to proteome expression in *Plasmodium falciparum*

Chee Sheng Ng, Ameya Sinha, Yaw Aniweh, Qianhui Nah, Indrakanti Ramesh Babu, Chen Gu, Yok Hian Chionh, Peter C. Dedon and Peter R. Preiser.

Review timeline:	Submission date:	27 th September 2017
	Editorial Decision:	12 th December 2017
	Revision received:	1 st June 2018
	Editorial Decision:	12 th July 2018
	Revision received:	23 rd July 2018
	Editorial Decision:	31 st July 2018
	Revision received:	9 th August 2018
	Accepted:	7 th September 2018

Editor: Thomas Lemberger.

Transaction Report:

1st Editorial Decision

12th December 2017

Dear Prof Preiser,

Thank you again for submitting your work to Molecular Systems Biology. We have now heard back from the three referees who agreed to evaluate your manuscript. As you will see from the reports below, the referees find the topic of your study of potential interest. They raise, however, substantial concerns on your work, which, I am afraid to say, preclude its publication in its present form.

Without repeating all the points noted in the reports below, the main issues refer to the need to 1) determine which proportion of a given tRNA species carry a modification (reviewer #1) and 2) to clarify/demonstrate the causal link between tRNA modification and alteration of translation efficiency (reviewer #3).

Please note that deposition of large-scale datasets in public resources is mandatory for publication in Molecular Systems Biology. We would therefore kindly ask you to provide the list of accession numbers to the datasets produced in this study in a formal Data Availability Section placed after Materials & Methods and that follows the model below:

#Data and software availability

The datasets and computer code produced in this study are available in the following databases:

- RNA-Seq data: Gene Expression Omnibus GSE46843
(<https://www.ncbi.nlm.nih.gov/geo/query/acc.cgi?acc=GSE46843>)
- Chip-Seq data: Gene Expression Omnibus GSE46748
(<https://www.ncbi.nlm.nih.gov/geo/query/acc.cgi?acc=GSE46748>)
- Protein interaction AP-MS data: PRIDE PXD000208

(<http://www.ebi.ac.uk/pride/archive/projects/PXD000208>)
- Imaging dataset: Image Data Resource doi:10.17867/10000101
(<http://doi.org/10.17867/10000101>)
- Modeling computer scripts: GitHub
(<https://github.com/SysBioChalmers/GECKO/releases/tag/v1.0>)
- data type: [full name of the resource] [accession number/identifier] ([doi or URL or
identifiers.org/DATABASE:ACCESSION])

If you feel you can satisfactorily deal with these points and those listed by the referees, you may wish to submit a revised version of your manuscript. Please attach a covering letter giving details of the way in which you have handled each of the points raised by the referees. A revised manuscript will be once again subject to review and you probably understand that we can give you no guarantee at this stage that the eventual outcome will be favorable.

REFeree REPORTS.

Reviewer #1:

My expertise is in the area of Plasmodium biology and genomics. My understanding of RNA modifications and mass spectrometry is more limited.

tRNA modifications have important regulatory functions, and while data on this subject are emerging rapidly from many areas of biology, little if any research has been done in Plasmodium, an organism which is of interest as a human pathogen, as a highly divergent eukaryote and possibly because of its extremely AT-rich genome. The existence and functions of tRNA modifications have not been explored systematically in this organism and a survey of such modifications is the first important contribution of this manuscript. Whether parasite nucleosides can be distinguished reliably from those of the host is absolutely critical for the interpretation of any temporal changes later on. I am fairly convinced the authors have done a good job here by including appropriate uninfected RBC controls and by looking specifically for human RNA contaminations (Fig. S1).

Following the initial survey, ribonucleoside modifications are quantified throughout the intraerythrocytic cycle. In Fig. 2 there appear to be some discrepancies between how data are presented in panels A and B, some of the categorisation seems arbitrary (see specific points below), and I wonder whether some of the more 'erratic' time courses are due to the detection limit on individual modification. Statistics are mentioned in the text but it is not clear how they were applied to the data. Overall, I am satisfied that different patterns exist for individual modifications. Whether these are likely to be meaningful is the subject of the subsequent analysis.

It is argued that changes in ribonucleoside modifications as described in Fig. 4 are of functional relevance. For this to be true, a significant proportion of a particular tRNA species would need to carry a modification, and this proportion would need to change. My understanding is that Fig. 2 tells us only how much the relative abundance of each nucleoside modification changes compared to all other modifications. This would not necessarily be very remarkable, for instance if the change only reflected the relative abundances of different tRNAs. Or perhaps only an insignificant proportion of a given tRNA is ever modified at any time point. What would be more important to know is whether the proportion of a particular tRNA carrying a specific modification that might affect its codon usage changes over time. For me this type of data would close an important gap in the argument. Figure 5 may contain part of the answer to my question, but is skipped over in the text and seems to be only for a single time point, which is not stated. Fig. 5 does, however, suggest to me temporal changes in modifications of a specific tRNA can be measured.

To link ribonucleoside modifications to translation, the authors establish a measure for translational efficiency and demonstrate statistically that late expressed proteins and early expressed proteins differ in their usage of relevant codons for some amino acids. I do believe these associations are striking. The specific genes that are characterised by extreme bias in codon usage and TE would be

of interest to malaria researchers and some of these could be discussed. Genome wide data underpinning all panels of Fig. 6 must be provided as supplemental information. Without this the analysis lacks transparency and reproducibility. The proteomics data underpinning Fig. 3 should also be shared and submitted to a public database.

The measure of TE somewhat debatable, of course. It would be good to see this discussed more critically. TE as determined here would be affected by protein stability, and presumable membrane or other proteins targeted to the micronemes late during schizogony would not turn over and lead to an overestimate of TE. Could this explain the observation of increased TE at the late time point? The readers are not shown the data to work this out by themselves (see previous paragraph).

The supplemental file is entitled 'Translational Control Mechanisms Governed by tRNA Modifications in *Plasmodium falciparum*'. This is a gross overstatement, which in the actual manuscript is toned down to 'tRNA epitranscriptomics and biased codon usage modulate translation in *Plasmodium falciparum*.' Even 'modulate' implies a causal link between tRNA modifications and TE, which the manuscript does not provide. The data are descriptive and establish a correlation, but there is not experimentation that could provide evidence for causation. This is not to say the work is not interesting or publishable in its current form, but the data must not be overinterpreted. This is also important because for practical purposes, codon usage does not appear to pose challenges for transgene expression in *Plasmodium*, so the actual effects may be limited.

If this work can be revised appropriately, this analysis will provide a solid rationale for the future identification and analysis of the *Plasmodium* enzymes involved in tRNA modifications, and their expression patterns. Gene knockout experiments may reveal any impact tRNA modifications have on parasite survival and virulence. Reporter constructs differing in codon usage would allow impact on TE to be measured in a controlled manner. I believe such additional validation experiments will not necessarily have to become part of the current manuscript, if issues above are resolved and the data are interpreted appropriately.

Specific points:

Abstract first line(s) missing.

P6 On the top of page 6 they say "m4Cm may thus be derived from either cytosolic or apicoplast tRNAs." Do they mean cytosolic or mitochondrial? Why do they rule out mitochondrial origin of a modification? Couldn't the ribonucleotides be modified following import into the mitochondria? This is not very clear.

P6 'Focusing our comparative analysis of tRNA modifications on the eukaryotes *S. cerevisiae* and *Homo sapiens* (Fig. 1C) revealed two (ms2t6A, m3U) unique to the parasite.' This is an unusual way of using 'revealing' and 'unique'. How can restricting one's view be revealing? Why don't they simply say two modifications were not shared with *Sc* and *Hs*? If they are unique to *Plasmodium*, that would be interesting, but then they should work that out.

P6 'The presence of three parasite-specific modifications (m4Cm, ms2t6A, m3U) raises the potential for targeting their biosynthetic pathways for drug development.' Better '...three modifications not shared with the human host...!'

P6 'modifications with one exception (ncm5Um): ncm5U, mcm5U, mcm5s2U, Ψ, s2U, Gm, Cm, I, and m5C'. Is the one in brackets the exception? Rephrase?

P6 'Parasite tRNA modifications reprogram across the IDC.' This is silly language. All that is shown is that they change in abundance. The grammar implies the modifications actively change some sort of program, of which there is no evidence.

Fig. 2 and legend. Panel A shows Log2 fold change and panel B just fold change but the two do not add up. For instance, from A, Am would seem to change between 2.6 and 0.4 fold, but this is not what B shows. How were the categories indicated by vertical double arrows decided? This seems arbitrary/ subjective. Why are genes in the third category unique if there are two members? The text says some of the changes were not statistically significant. Which were and how was it worked out?

The graphs in B are presented as if they were representative of the three categories although they are clearly not.

In Fig. 6 E and F why label each data point?

Fig. S2 and elsewhere: Malaria Parasite Metabolic (MPM) and GO pathway enrichment analysis lacks appropriate references as to what these are.

Other phrases that may be revised:

'and significant delay in transcript and protein levels'

'However, these gene-specific mechanisms do not reveal general, systems-level mechanisms'

'in which its nuclear genome containing only one gene copy per tRNA isoacceptor'

'we first identified and quantified spectrum of modified ribonucleosides'

P10 'Translational efficiency accounts for codon-biased up-regulation of proteins late IDC.' Missing words.

Reviewer #2:

The manuscript entitled "tRNA epitranscriptomics and biased codon usage modulate translation in *Plasmodium falciparum*" by Chee Sheng Ng and colleagues investigates changes in tRNA modifications and translational efficiency in the malaria parasite intra erythrocytic developmental cycle (IDC) using systems-level proteomic, genomic and epitranscriptomic analyses. Results of this analysis provide new insights into tRNA maturation and reveal a coordinated system of tRNA modification reprogramming coupled to codon-biased translation to fine-tune parasite protein needs across the IDC. Overall, the topic of the manuscript is highly relevant to understanding fundamentals of malaria parasite biology. It is written clearly and logically, and should be of general interest to the pathogens community. Most importantly, result of this work has a huge implication for the identification of new targets for development of antimalarial drugs.

I have however several comments that should be addressed before the manuscript is accepted for publication.

- 1- The abstract may have been truncated! It seems that the first few sentences are missing
- 2- When the authors cite published works on mechanisms regulating post-transcriptional control in *P. falciparum* such as regulation of translation by untranslated regions (UTR), the authors should have included the first paper describing this phenomenon at the genome-wide level in *Plasmodium* (Bunnik et al., *Genome Biology* 2013).
- 3- I don't really understand why the authors performed a new proteomics analysis across the IDC? Several high-quality proteomics data sets have been published and could have been used/incorporated into the data analyses. If the proteomics data generated previously were inappropriate, the authors should explained why.
- 4- The authors used old published RNA-seq data (Otto et al., 2010) to calculate the index of translational efficiency (TE) for each protein analyzed in the parasite. More recent RNA-seq data with increased sequence depth have been published and should have been considered.
- 5- Ribosome footprinting data set has been published a few years ago in *P. falciparum* and should have been considered/integrated in the calculation of the translational efficiency (Caro F et al., *Elife* 2014). If the Ribosome foot printing data set generated previously were inappropriate for this analysis, the authors should explained why.

Reviewer #3:

This manuscript contains a careful analysis of tRNA modifications in the asexual blood stages of

Plasmodium falciparum, and a detailed examination of their temporal abundance changes through the blood stages. This is important research and its publication will allow others to further characterise the purpose and importance of these modifications.

The authors also present a lengthy analysis that draws correlations between abundance of specific modified nucleosides and the expression of different families of proteins by the parasite during the asexual cycle. I found this section interesting, and some plausible hypotheses are raised that could be tested, but this section is still quite speculative and none of the observational correlations are followed up by testing the falsifiable hypotheses.

In general, this is an interesting observational paper, where correlations between datasets are used to predict a model of gene regulation, but those correlations are not clearly explained, not clearly depicted, and no complete mechanism is proposed or tested to substantiate such a model. The authors show unsurprisingly that most tRNA modifications become more abundant as bulk protein translation increases during the life cycle, although two modifications become less abundant in late stages. The changes in a handful of others are noisy or not strongly correlated. The exception of the two modifications that move in a contrary direction to the bulk of others that generally increase does not at first glance appear to contain sufficient signal to build much of a model from, but the authors infer a model that suggests that the tRNA modifications allow changes in translating particular codons, and that this results in changes in translational efficiency. This could be tested by modifying or eliminating the expression of genes required for those tRNA modifications - for example, one from the group that is upregulated in late stages, one from the group upregulated in early stages. Some have relatively simple single-enzyme pathways, so gene disruptions could be chosen that modify a single gene could, and several of these knockouts appear to be perfectly viable based on high-throughput *P. berghei* PlasmoGem knockouts. Those parasite could then be assayed to test whether those tRNA changes result in modulations in the translational efficiency (or protein abundance) of the proteins hypothesised to be influenced by those modified tRNAs.

I feel that the tRNA modification data and the protein abundance data are worthwhile datasets that would be appropriately published in a parasitology-discipline-specific journal, but do not constitute a robust systems biology model appropriate to publication in *Molecular Systems Biology*.

Additional points

The authors describe the protein abundance changes that they observe as differences in translational efficiency. It is quite unclear to me that this is what has been measured. The authors measure protein abundance, but no attempt has been made, as far as I could see, to normalise this for protein turnover. Therefore a valid alternative hypothesis is that all proteins are equally translated, and differences in abundance are merely a product of differences in rates of protein degradation. Although the authors do normalise for

The authors have previously published quantitative proteomics of *Plasmodium* based on other methods. How does the iTRAQ quantification compare to these earlier attempts - does it appear to be more accurate assessment of quantification based on any independent analysis (e.g. western blots of representative proteins)?

The means used to demonstrate correlation using PCA seems a non-intuitive method to demonstrate relationships between proteome changes and tRNA modifications. Although these plots suggest that relationships exist, they don't show what the nature of this relationship is, and I missed a figure that showed this in a more quantitative way over the proteome, or separated for individual modifications.

1st Revision - authors' response

1st June 2018

Response to Reviewer Comments for *Molecular Systems Biology* manuscript MSB17-8009, "tRNA epitranscriptomics and biased codon usage modulate translation in *Plasmodium falciparum*," by Ng et al.

We thank the reviewers and editors for a very thoughtful evaluation of this manuscript and we greatly appreciate the overall positive response to our work. We have now responded to all of the reviewer comments in the revised manuscript (changes in red font), with the responses noted in

detail here. The scholarship and quality of the manuscript is significantly improved thanks to the thorough review.

Reviewer #1

- My expertise is in the area of Plasmodium biology and genomics. My understanding of RNA modifications and mass spectrometry is more limited. tRNA modifications have important regulatory functions, and while data on this subject are emerging rapidly from many areas of biology, little if any research has been done in Plasmodium, an organism which is of interest as a human pathogen, as a highly divergent eukaryote and possibly because of its extremely AT-rich genome. The existence and functions of tRNA modifications have not been explored systematically in this organism and a survey of such modifications is the first important contribution of this manuscript. Whether parasite nucleosides can be distinguished reliably from those of the host is absolutely critical for the interpretation of any temporal changes later on. I am fairly convinced the authors have done a good job here by including appropriate uninfected RBC controls and by looking specifically for human RNA contaminations (Fig. S1).*

We appreciate the sentiment that the work detailed in the paper is of interest to the field of *Plasmodium* biology and have made the revisions throughout the manuscript as specified by the reviewer where indicated below.

- Following the initial survey, ribonucleoside modifications are quantified throughout the intraerythrocytic cycle. In Fig. 2 there appear to be some discrepancies between how data are presented in panels A and B, some of the categorisation seems arbitrary (see specific points below), and I wonder whether some of the more 'erratic' time courses are due to the detection limit on individual modification. Statistics are mentioned in the text but it is not clear how they were applied to the data. Overall, I am satisfied that different patterns exist for individual modifications. Whether these are likely to be meaningful is the subject of the subsequent analysis.*

We apologize for the confusion in Figure 2. The heat map in Panel A is based on \log_2 transformation of the fold-change ratios that were depicted in the graphs in the original Panel B. Also, the color scale for the heat map in original Panel A was confusing in noting \log_2 transformation of the indicated fold-change values. We have revised Figure 2 to correct the color scale error in Panel A and to use $\log_2(\text{fold-change})$ values for the y-axes of graphs in Panel B. The data in Panels A and B are now directly comparable.

The reviewer raises concerns about the role of signal intensity in the patterns of changes in modification levels in Figure 2, with modifications not following the time-dependent upward trend seen for most modifications labeled as “erratic”. We assume that the reviewer means that the erratic modifications have low signal intensities and thus reflect noise rather than reproducible and biological meaningful signals. We greatly appreciate this concern for the quality of the analytical data for the tRNA modifications and we erred by not including the normalized signal intensities for the modifications at each time point, which demonstrate the high precision and accuracy of the data. In addition to the fold-change data for the modifications, we have now included the normalized signal intensities for the three replicate analyses for each modified nucleoside at each time point in Table S6. For two of the erratic modifications, N⁶-methyladenosine (m6A) and pseudouridine (Y), the variance in the triplicate data is similar to other modifications at <10%, which reflects the highly precise nature of the LC-MS quantitative analysis. For one erratic modification, N⁶,N⁶-dimethyladenosine (m66A), the variance is large at ~50%, which is consistent with the low signal intensities for m66A (see Table S6) and with the reviewer’s comment. We have also revised the manuscript to comment on the analytical characteristics of the modified nucleosides. The data for each modification are, with few exceptions, well above the limit of detection and highly precise (i.e., low variance among samples).

We respectfully disagree with the reviewer’s assertion that the assignment of modifications into the three categories is arbitrary. To the contrary, the self-organizing map arising from hierarchical clustering has arranged the modifications according to three distinct patterns of covariance, which is the basis for multivariate statistical analysis and is driven by the high precision of the data. The heat

map merely visualizes what the mathematics determined as two major categories for the behavior of the modifications, as we noted in the text: those that rise synchronously along the ring-to-schizost time course and those that do not follow this behavior (“asynchronous”). The multivariate statistical analysis further partitions the asynchronous modifications into two classes: two modifications that showed behavior opposite the synchronous modifications (Am, s2U; decreasing across the RBC lifecycle), and three of the asynchronous modifications that showed the “erratic” behavior noted by the reviewer. So the assignment of three categories of behavior is well justified and not arbitrary, with the mechanisms driving these three behaviors awaiting further characterization in future studies.

We apologize for not clearly describing the statistical treatments applied to the data in Figure 2, which has now been rectified in the revised manuscript. As described in the methods section of the original manuscript (page 30, line 1), data normalization for hierarchical clustering analysis occurs in two stages. In the first stage, raw mass spectrometer signals are processed as fold-change values as follows. First, the mass spec signal intensity for each ribonucleoside is normalized by dividing the raw peak area for the ribonucleoside by the UV absorbance peak areas (in-line diode array detector) for the four canonical ribonucleosides. This adjusts for variations in total tRNA in the sample. Second, the normalized signal intensity of each ribonucleoside was further normalized against an internal standard ($[^{15}\text{N}]\text{-2}'\text{-deoxyadenosine}$) spiked into each sample during RNA digestion to adjust for day-to-day fluctuation in the response of the mass spectrometer. Third, we calculated lifecycle-dependent changes in the quantities of modified ribonucleosides as fold-changes using the normalized values for each time point relative to an arbitrary average over the six time points. In the second stage of analysis, fold-change values for each biological replicate were averaged (arithmetic mean) as a single value for representation in the heat map presented in Fig. 2a.

The statistical analysis for the three modifications in Panel B of the original Fig 2 was done by comparing the differences between the highest value (denoted as “peak”) and the lowest value (denoted as “trough”) of the modification fold-change levels across the 6 time points. The highest and lowest fold-change values were subjected to two-tailed Student’s t-test: NS, not significant; $P < 0.05$, *; $P < 0.01$, **. A summary table of the tRNA modification levels and P-values has now been added in Table S6, as noted earlier.

- It is argued that changes in ribonucleoside modifications as described in Fig. 4 are of functional relevance. For this to be true, a significant proportion of a particular tRNA species would need to carry a modification, and this proportion would need to change. My understanding is that Fig. 2 tells us only how much the relative abundance of each nucleoside modification changes compared to all other modifications. This would not necessarily be very remarkable, for instance if the change only reflected the relative abundances of different tRNAs. Or perhaps only an insignificant proportion of a given tRNA is ever modified at any time point. What would be more important to know is whether the proportion of a particular tRNA carrying a specific modification that might affect its codon usage changes over time. For me this type of data would close an important gap in the argument. Figure 5 may contain part of the answer to my question, but is skipped over in the text and seems to be only for a single time point, which is not stated. Fig. 5 does, however, suggest to me temporal changes in modifications of a specific tRNA can be measured.*

The reviewer raises an important issue about differentiating between population-level changes in modified ribonucleosides and tRNA copy numbers. To address this issue, we have now performed experiments to quantify changes in tRNA copy numbers and wobble modifications for three tRNA isoacceptors (tRNA^{GlnCAA}, tRNA^{GluGAA}, tRNA^{GlyGGA}), two of which are associated with codons found to be enriched in genes for up-regulated proteins during the IDC (tRNA^{GluGAA}, tRNA^{GlyGGA}). The experiment involved isolation of total tRNA from ring (10-16H), trophozoite (28-34H), and schizont (38-44H) stages of the IDC, followed by RNase T1 cleavage to generate unique oligonucleotides from each isoacceptor. The oligos were then quantified and sequenced by mass spectrometry to define the modification changes. As shown in new Figure S4 (also shown in the adjacent graph), the number of copies of the three tRNA isoacceptors did not change significantly across the IDC, which is consistent with tRNA microarray analysis of the *Plasmodium* lifecycle by Rovira-Graells *et al.* (2012). This suggests a relatively unchanging pool of tRNA copy numbers at least for the glutamine, glutamic acid and glycine isoacceptors. MRM method to detect for 2 of the most abundant ions (w and y) for the oligonucleotide representing the anticodon stem loop based on the fragmentation pattern obtained on the qToF (Fig 5C,5F, Sup Fig S3C). We have included this experiment as part of the manuscript (P9L20 P10L1 and Fig 5G 5H 5I) and present summary data for your consideration.

The situation is different for the wobble modifications on these three tRNAs. As shown in the figure below (new Figure 5 panels G-I), the fraction of each tRNA isoacceptor modified with either mcm5U or mcm5s2U increases across the IDC, which is consistent with both the population increase in these modifications and the up-regulation of proteins from genes enriched with the cognate codons of these tRNAs.

These results support a model in which changes in tRNA modifications influence the translation of codon-biased mRNAs during the parasite RBC developmental cycle. We have revised the manuscript with these studies: page 10, lines 1-20; Fig S4; Fig 5, panels G-I.

Figure 5G-I: IDC-dependent remodeling of the tRNA^{Gly}, tRNA^{Glu}, and tRNA^{Gln}. Ratios of modified oligonucleotides from the anticodon stem-loop of (G) Gln^{CAA} (H) Glu^{GAA} (I) Gly^{GGA} isoacceptors expressed as percentages of total tRNA isoacceptor levels.

- To link ribonucleoside modifications to translation, the authors establish a measure for translational efficiency and demonstrate statistically that late expressed proteins and early expressed proteins differ in their usage of relevant codons for some amino acids. I do believe these associations are striking. The specific genes that are characterized by extreme bias in codon usage and TE would be of interest to malaria researchers and some of these could be discussed. Genome wide data underpinning all panels of Fig. 6 must be provided as supplemental information. Without this the analysis lacks transparency and reproducibility. The proteomics data underpinning Fig. 3 should also be shared and submitted to a public database.*

We apologize for this oversight and have now included supplemental tables that contain data for the genome wide codon usage (Table S9) and for the translational efficiency calculations (Table S8) (revised manuscript page 11, line 4). We have uploaded the processed proteomics data from

Scaffold as Table S7. We have also uploaded all of the proteomics data as a submission to the CHROUS database as requested by the reviewer: Project No. 1443: Plasmodium IDC iTRAQ; this is noted in the revised manuscript.

- *The measure of TE somewhat debatable, of course. It would be good to see this discussed more critically. TE as determined here would be affected by protein stability, and presumable membrane or other proteins targeted to the micronemes late during schizogony would not turn over and lead to an overestimate of TE. Could this explain the observation of increased TE at the late time point? The readers are not shown the data to work this out by themselves (see previous paragraph).*

We agree with the reviewer that measuring translational efficiency is complicated by the number of variables involved in RNA and protein stability, particularly since we have not factored in mRNA half-life, ribosome loading, proteosomal degradation, and protein turnover. The TE metric used in this paper is similar to that used by other groups, such as Schwanhäuser *et al.* (2011), using mRNA expression and protein abundance in calculating TE. We chose not to include published 'omic data in the TE calculations given the confounders introduced by including data sets obtained at different time points and different growth conditions. As noted in the revised manuscript (page 31, line 10), we quantified an arbitrary TE index defined as the ratio of the fold-change value for a protein to the fold-change value for the corresponding mRNA at a specific time point. We apologize for not including the details of our approach to TE calculation and the resulting data in the manuscript and we have now provided the relevant figures underpinning the graphs depicted in the Table S8. To the reviewer's point about late-stage TE overestimation, we point out that our proteomics data reflects mainly soluble/cytosolic proteins, such that membrane proteins are likely underrepresented and do not contribute significantly to the TE calculations.

- *The supplemental file is entitled 'Translational Control Mechanisms Governed by tRNA Modifications in Plasmodium falciparum'. This is a gross overstatement, which in the actual manuscript is toned down to 'tRNA epitranscriptomics and biased codon usage modulate translation in Plasmodium falciparum.' Even 'modulate' implies a causal link between tRNA modifications and TE, which the manuscript does not provide. The data are descriptive and establish a correlation, but there is not experimentation that could provide evidence for causation. This is not to say the work is not interesting or publishable in its current form, but the data must not be overinterpreted. This is also important because for practical purposes, codon usage does not appear to pose challenges for transgene expression in Plasmodium, so the actual effects may be limited.*

If this work can be revised appropriately, this analysis will provide a solid rationale for the future identification and analysis of the Plasmodium enzymes involved in tRNA modifications, and their expression patterns. Gene knockout experiments may reveal any impact tRNA modifications have on parasite survival and virulence. Reporter constructs differing in codon usage would allow impact on TE to be measured in a controlled manner. I believe such additional validation experiments will not necessarily have to become part of the current manuscript, if issues above are resolved and the data are interpreted appropriately.

We apologize for the different titles in the main text and the supplementary information. We agree that the supplemental file title is an overstatement of the conclusions reached in the manuscript and we had indeed toned it down with the present title. The reviewer's point about association versus causation is well taken. Here we point out that evidence for condition-specific modification changes in the tRNAs that read the codons enriched in genes that are specifically up- and down-regulated during the condition is indeed evidence of a causative mechanism and not simply an association. This is especially true in light of our many published observations of such causative links in yeast and bacteria (cited in the manuscript; PMIDs 26253965, 25772370, 21187895, 22760636, 27834374, 26670883, 27245397). We have nonetheless changed the title to further soften the causation conclusion and have revised the manuscript to avoid over-reaching conclusions. As steps toward further proof of causation, we appreciate the reviewer's suggestions for follow-up experiments, including codon-biased reporters and genetic approaches for tRNA modifying enzymes. We also appreciate the reviewer's broad view that such experiments are beyond the scope of the present work.

Specific points:

- *Abstract first line(s) missing.*

We apologize for what must have been an error during PDF conversion. We have rectified this problem in the newly uploaded revised manuscript.

- *P6 On the top of page 6 they say "m4Cm may thus be derived from either cytosolic or apicoplast tRNAs." Do they mean cytosolic or mitochondrial? Why do they rule out mitochondrial origin of a modification? Couldn't the ribonucleotides be modified following import into the mitochondria? This is not very clear.*

We thank the reviewer for the comment and wish to clarify the situation with mitochondrial genes in *Plasmodium*. The small (6 kb) *Plasmodium* mitochondrial genome does not possess the requisite enzymes to carry out any aspect of the m4Cm modification. <https://www.ncbi.nlm.nih.gov/nucleotide/AY282930>. It is currently known to code only for three proteins (cytochrome oxidase subunits 1 and 3, and cytochrome B). Thus it is highly unlikely that the tRNA is modified by mitochondrially encoded enzymes post-import, unless the enzyme is imported as well. In addition, Suzuki *et al.* (2014) have proposed the most likely modifications that occur in mammalian mitochondrial tRNA and m4Cm is not among them. The most probable hypothesis is that these modified tRNA are either derived from the apicoplast or are present in the cytosol and are not of mitochondrial origin. The manuscript (page 5, lines 25 and 26; page 6, line 2 and line 5) has been revised for the benefit of the reader and to avoid any confusion.

- *P6 'Focusing our comparative analysis of tRNA modifications on the eukaryotes *S. cerevisiae* and *Homo sapiens* (Fig. 1C) revealed two (ms2t6A, m3U) unique to the parasite.' This is an unusual way of using 'revealing' and 'unique'. How can restricting one's view be revealing? Why don't they simply say two modifications were not shared with *Sc* and *Hs*? If they are unique to *Plasmodium*, that would be interesting, but then they should work that out.*

We appreciate the distinction and clarity offered by the reviewer and have revised the manuscript to simplify the statement (page 6, line 6).

- *P6 'The presence of three parasite-specific modifications (m4Cm, ms2t6A, m3U) raises the potential for targeting their biosynthetic pathways for drug development.' Better '...three modifications not shared with the human host...'*

We have revised the manuscript with this comment (page 6, line 8).

- *P6 'modifications with one exception (ncm5Um): ncm5U, mcm5U, mcm5s2U, Ψ, s2U, Gm, Cm, I, and m5C'. Is the one in brackets the exception? Rephrase?*

The bracketed modification is the exception and have revised the statement to avoid misinterpretation (page 6, line 12).

- *P6 'Parasite tRNA modifications reprogram across the IDC.' This is silly language. All that is shown is that they change in abundance. The grammar implies the modifications actively change some sort of program, of which there is no evidence*

We respectfully disagree with the reviewer's comment that reprogramming of tRNA modifications is "silly language". Our published studies in yeast and bacteria (PMIDs 26253965, 25772370, 21187895, 22760636, 27834374, 26670883, 27245397) conclusively and mechanistically demonstrate a program of specific changes in the levels of tRNA modifications on specific isoacceptors, which coordinates with an alternative genetic code of synonymous codon usage to alter translation of families of genes required to change cell phenotype. Where we erred in the present manuscript is to use this same language for the observations made in the *Plasmodium* red blood cell life cycle, for which such mechanistic certainty has not been established, as the reviewer

pointed out earlier. We have revised the manuscript to remove “reprogramming” to describe the observed tRNA modifications, except to propose this as a model to be tested in future studies to conclusively establish such a reprogramming mechanism.

- *Fig. 2 and legend. Panel A shows Log2 fold change and panel B just fold change but the two do not add up. For instance, from A, Am would seem to change between 2.6 and 0.4 fold, but this is not what B shows. How were the categories indicated by vertical double arrows decided? This seems arbitrary/ subjective. Why are genes in the third category unique if there are two members? The text says some of the changes were not statistically significant. Which were and how was it worked out? The graphs in B are presented as if they were representative of the three categories although they are clearly not.*

We apologize for the confusion in Figure 2. As noted in detail earlier, we have now revised Figure 2 for consistency among the panels and we articulated the mathematical basis for organizing the modifications behavior into three distinct patterns. Please see the earlier comments.

- *In Fig. 6 E and F why label each data point?*

The reviewer notes a redundancy in the scores plots (panels E, F), with both color and text labels for the proteins with high and low translational efficiency at 40 and 46 hours of the parasite lifecycle. We have revised the figure panels to remove the text labels, which allows clearer visualization of the data points. The text labels for the loadings plots (G, H) are essential to distinguish the various codons.

- *Fig. S2 and elsewhere: Malaria Parasite Metabolic (MPM) and GO pathway enrichment analysis lacks appropriate references as to what these are.*

We have provided descriptions for both MPM and the GO pathway enrichment analysis for the benefit of the reader.

- *Other phrases that may be revised:
'and significant delay in transcript and protein levels'
'However, these gene-specific mechanisms do not reveal general, systems-level mechanisms'
'in which its nuclear genome containing only one gene copy per tRNA isoacceptor'
'we first identified and quantified spectrum of modified ribonucleosides'
P10 'Translational efficiency accounts for codon-biased up-regulation of proteins late IDC.' Missing words.*

We have revised the manuscript appropriately to omit and/or rephrase these points and thank the reviewer for improving the readability of the manuscript.

Reviewer #2

- *The manuscript entitled "tRNA epitranscriptomics and biased codon usage modulate translation in Plasmodium falciparum" by Chee Sheng Ng and colleagues investigates changes in tRNA modifications and translational efficiency in the malaria parasite intra erythrocytic developmental cycle (IDC) using systems-level proteomic, genomic and epitranscriptomic analyses. Results of this analysis provide new insights into tRNA maturation and reveal a coordinated system of tRNA modification reprogramming coupled to codon-biased translation to fine-tune parasite protein needs across the IDC. Overall, the topic of the manuscript is highly relevant to understanding fundamentals of malaria parasite biology. It is written clearly and logically, and should be of general interest to the pathogens community. Most importantly, result of this work has a huge implication for the identification of new targets for development of antimalarial drugs. I have however several comments that should be addressed before the manuscript is accepted for publication.*

We thank reviewer #2 for the interest in our work and for pointing out several mistakes and areas for improvement. We made revised the manuscript based on the comments detailed below.

- *The abstract may have been truncated! It seems that the first few sentences are missing*

We have corrected this error as noted in the response to Reviewer #1.

- *When the authors cite published works on mechanisms regulating post-transcriptional control in P. falciparum such as regulation of translation by untranslated regions (UTR), the authors should have included the first paper describing this phenomenon at the genome-wide level in Plasmodium (Bunnik et al., Genome Biology 2013).*

We apologize for the oversight and have included the reference in our manuscript text. We have also added a line to the discussion in the manuscript (page 13, line 10) to discuss the polysomal profiling carried out by Bunnik et al. (2013), which shows that >50% of the genes in the asexual stage of *Plasmodium* exhibited a shift in the protein abundance as compared to transcript abundance, with some genes showing a delay of more than 18 hours. Based on this observation, coupled with the lack of transcriptional control mechanisms, they suggest a just-in-time translation of proteins, which is in line with our proposed model. Not mutually exclusive, Bunnik et al. point to translational repression at uORFs, whereas our identifies a role for tRNA modifications in translation. We thank the reviewer for pointing out this relevant publication, which emphasizes the need to investigate alternative mechanisms of translational control.

- *I don't really understand why the authors performed a new proteomics analysis across the IDC? Several high-quality proteomics data sets have been published and could have been used/incorporated into the data analyses. If the proteomics data generated previously were inappropriate, the authors should explained why.*

We appreciate the reviewer's concern for the risk of redundancy in the expensive and time-consuming proteomics analyses. There are four critical proteomic datasets that we assessed:

- Florens et al. (2002) -2400 proteins but only sporozoites, merozoites, trophozoites and gametocytes; label-free quantitation.
- Le Roch et al. (2004) - Sporozoites, gametocytes, trophozoites, and merozoites. 2900 proteins in at least one of the analyses, many with in appropriate single peptide identification.
- Foth et al. (2008) – 623 proteins using DIGE at 34, 38, 42, and 46 hours post-infection (hpi).
- Foth et al. (2011) – 125 *Plasmodium* proteins using DIGE every 4 hours.

These datasets did not provide sufficient quantitative data at the same time resolution required for our analysis. These proteomics datasets were carried out either using different strains or *Plasmodium* species or in response to drug treatments at a single timepoint. These problems obviated the use of these datasets in our studies. Our proteomics dataset identifies 2100 proteins that were quantitatively identified at all 6 time points, which represents the best coverage of the *Plasmodium* asexual stage proteome to date. We have included an additional point in the manuscript

(page 7, line 21) to clarify these issues.

- *The authors used old published RNA-seq data (Otto et al., 2010) to calculate the index of translational efficiency (TE) for each protein analyzed in the parasite. More recent RNA-seq data with increased sequence depth have been published and should have been considered.*

We chose RNA-seq data from Otto et al. (2010) as it not only matches the time points of our proteomic data but also measures steady-state RNA levels, which we feel is most relevant for steady-state translational activity. We are indeed aware of the Lu et al. (2017) dataset, which we have acknowledged in the manuscript (page 11, line 18). However, the Lu et al. dataset measures nascent RNA transcripts rather than the steady-state RNA levels measured in the Otto et al. study. As translational activity would be a function of the available transcript at a given point in time we feel that the steady-state RNA data set is more appropriate for our studies. We have included an additional point in the manuscript (page 11, line 14) to clarify this issue.

- *Ribosome footprinting data set has been published a few years ago in *P. falciparum* and should have been considered/integrated in the calculation of the translational efficiency (Caro F et al., Elife 2014). If the Ribosome foot printing data set generated previously were inappropriate for this analysis, the authors should explained why.*

We appreciate the reviewer's rigor in evaluating parasite 'omic studies. We are indeed familiar with the Caro F et al. (2014) ribosome footprinting in *Plasmodium*. We chose not to consider this study in our translation efficiency calculations since the study differed in the *P. falciparum* strain used (W2) and in the time points assayed (11, 21, 31, and 45 hr). While ribosome profiling would add an important dimension to a translation efficiency calculation, the use of a dataset obtained under very different conditions could produce artifacts. We have noted this issue in the revised manuscript (page 11, line 14).

Reviewer #3

- *This manuscript contains a careful analysis of tRNA modifications in the asexual blood stages of Plasmodium falciparum, and a detailed examination of their temporal abundance changes through the blood stages. This is important research and its publication will allow others to further characterize the purpose and importance of these modifications. The authors also present a lengthy analysis that draws correlations between abundance of specific modified nucleosides and the expression of different families of proteins by the parasite during the asexual cycle. I found this section interesting, and some plausible hypotheses are raised that could be tested, but this section is still quite speculative and none of the observational correlations are followed up by testing the falsifiable hypotheses.*
- *In general, this is an interesting observational paper, where correlations between datasets are used to predict a model of gene regulation, but those correlations are not clearly explained, not clearly depicted, and no complete mechanism is proposed or tested to substantiate such a model. The authors show unsurprisingly that most tRNA modifications become more abundant as bulk protein translation increases during the life cycle, although two modifications become less abundant in late stages. The changes in a handful of others are noisy or not strongly correlated. The exception of the two modifications that move in a contrary direction to the bulk of others that generally increase does not at first glance appear to contain sufficient signal to build much of a model from, but the authors infer a model that suggests that the tRNA modifications allow changes in translating particular codons, and that this results in changes in translational efficiency. This could be tested by modifying or eliminating the expression of genes required for those tRNA modifications - for example, one from the group that is upregulated in late stages, one from the group upregulated in early stages. Some have relatively simple single-enzyme pathways, so gene disruptions could be chosen that modify a single gene could, and several of these knockouts appear to be perfectly viable based on high-throughput P. berghei PlasmoGem knockouts. Those parasites could then be assayed to test whether those tRNA changes result in modulations in the translational efficiency (or protein abundance) of the proteins hypothesised to be influenced by those modified tRNAs.*

We thank the reviewer for the overall positive response to the quality of the data and to aspects of the proposed models. However, we respectfully disagree that the work presented in the manuscript is observational and lacking a clearly defined model. Throughout the manuscript, we develop a mechanistic model of tRNA modification changes and codon-biased translation based on quantitative analysis of the interplay among multiple molecular and genomic systems – transcriptome, ribonucleome, epitranscriptome, protein and genome – across the *Plasmodium* RBC life cycle. The data supporting each facet of the model are discussed thoroughly and the facets needing additional study are clearly noted. For example, we agree with the reviewer's comment that the increased levels of most modifications along the developmental cycle are consistent with the associated up-regulation in a variety of molecular processes as the parasite matures and we discussed this in the original manuscript. We respectfully point out, however, that reviewer appears to have missed our discussion of the idea of two parallel processes occurring in the modification changes: tRNA maturation and stage-dependent changes in modifications that promote codon-biased translation. As noted earlier, we have now revised the manuscript with data showing stage-dependent increases in specific wobble modifications on tRNA isoacceptors that read the codons enriched in up-regulated proteins at each stage. While these observations are consistent with the model, we agree with the reviewer that additional genetic studies are needed to firmly establish a causal link between the specific modifications and codon-biased translation. However, these studies are well beyond the scope of the present work, in which we develop a model based on an extensive analysis of multi-omic datasets.

- *I feel that the tRNA modification data and the protein abundance data are worthwhile datasets that would be appropriately published in a parasitology-discipline-specific journal, but do not constitute a robust systems biology model appropriate to publication in Molecular Systems Biology.*

We strongly disagree with the reviewer's conclusion that our work does not constitute a robust systems biology model. To the contrary, as noted earlier, we developed a mechanistic model of tRNA modification changes and codon-biased translation based on quantitative analysis of the interplay among multiple molecular and genomic systems – transcriptome, ribonucleome, epitranscriptome, protein and genome – across the *Plasmodium* RBC life cycle. This work is well suited to *Molecular Systems Biology*. The model is strongly supported by the data and by existing knowledge of limited transcriptional regulatory mechanisms in the parasite, and leads to a rich set of testable hypotheses about the relationship of this translational model to other systems affecting parasite development.

Additional points

- *The authors describe the protein abundance changes that they observe as differences in translational efficiency. It is quite unclear to me that this is what has been measured.*

As noted in the response to Reviewer #1, the translational efficiency metric used here is similar to that used by other groups, such as Schwanhäuser *et al.* (2011), using mRNA expression and protein abundance in calculating TE. We chose not to include other published 'omic data in the TE calculations given the confounders introduced by including data sets obtained at different time points and different growth conditions. As noted in the revised manuscript (page 31, line 10), we quantified an arbitrary TE index defined as the ratio of the fold-change value for a protein to the fold-change value for the corresponding mRNA at a specific time point. We apologize for not including the details of our approach to TE calculation and the resulting data in the manuscript and we have now provided the relevant figures underpinning the graphs depicted in the Table S8. To the reviewer's point about late-stage TE overestimation, we point out that our proteomics data reflects mainly soluble/cytosolic proteins, such that membrane proteins are likely underrepresented and do not contribute significantly to the TE calculations.

- *The authors measure protein abundance, but no attempt has been made, as far as I could see, to normalise this for protein turnover. Therefore a valid alternative hypothesis is that all proteins are equally translated, and differences in abundance are merely a product of differences in rates of protein degradation. Although the authors do normalise for*

The reviewer raises an important issue: the role of protein degradation in the full cycle of gene expression. Here we point out that protein degradation cannot account for our observation of unique codon-biased translation at each stage of parasite development. The codon information is lost in protein sequence and structure. While codon usage patterns in genes cannot influence protein degradation, we agree with the reviewer that protein turnover likely contributes to the steady-state level of proteins to some extent, which has implications for the complexity of our translation model. This will be an important problem for future studies. We have revised the manuscript (page 11, line 18) to include a discussion of the potential contributions of protein turnover.

- *The authors have previously published quantitative proteomics of Plasmodium based on other methods. How does the iTRAQ quantification compare to these earlier attempts - does it appear to be more accurate assessment of quantification based on any independent analysis (e.g. western blots of representative proteins)?*

The reviewer's question is an important one in light of the different proteomics datasets available in the *Plasmodium* literature; this point was discussed earlier in the response to Reviewer #2. iTRAQ, TMT and other isobaric tagging methods provide the most robust and comprehensive proteomics data, and the methods are well established and firmly validated in hundreds of publications. We have similarly validated iTRAQ proteomics in our published study in bacteria (PMCID: PMC5114619), using manual validation against published mass spectrometric proteomics and Western blot data sets. iTRAQ proteomics provides greater depth of coverage (thousands compared to hundreds of proteins), sensitivity, dynamic range, precision and accuracy than 2-D gels, while it is similar to SILAC proteomics. In order to validate the dataset used in the manuscript, we manually evaluated a subset of proteins and compared the data to literature values. For example, Counihan *et al.* (2017) used Western blots to quantify the behavior of the apical membrane antigen 1 by across the lifecycle (left side; note: figure has been cropped), which compares well with our proteomics

data for the same protein averaged for three independent biological replicates (right side). We generated this example for the purpose of this response to reviewers but we have not included in the manuscript since iTRAQ proteomics is now an accepted and well validated method.

<Western blot from Counihan et al. (2017); for copyright reasons, the image is not displayed in this Review Process File>

- *The means used to demonstrate correlation using PCA seems a non-intuitive method to demonstrate relationships between proteome changes and tRNA modifications. Although these plots suggest that relationships exist, they don't show what the nature of this relationship is, and I missed a figure that showed this in a more quantitative way over the proteome, or separated for individual modifications.*

Principal component analysis (PCA) is a widely used statistical tool to identify the most significant covariance among data in large systems-level datasets. It is one of many multivariate statistical tools that include hierarchical clustering analysis, as in the ribonucleoside heat map in Figure 2. The PCA analysis in Figure 4 was performed to quantify the association between codon usage and protein up- and down-regulation at each time point of the parasite RBC developmental cycle, while that in Figure 6 expands the covariation analysis to include translational efficiency (TE). Both figures show that highly up-regulated proteins derive from genes enriched with specific synonymous codons. The reviewer is correct in noting that the PCA analysis shows association but does not reveal mechanisms. We propose that the mechanism underlying the association of codon biases and translational efficiency is that the biased codons are read by tRNAs shown to possess increasing levels of wobble modifications across the IDC. This is illustrated for three tRNA isoacceptors in new Figure 5: these tRNAs read codons enriched in genes for up-regulated proteins and they possess wobble modifications found to increase across the IDC. While we have not established a causal link to prove this model, the point of the manuscript is that we used systems-level analyses to identify patterns of behavior that are consistent with this model, a model that was established unequivocally in yeast and bacteria.

2nd Editorial Decision

12th July 2018

Dear Prof. Preiser,

Thank you again for submitting your revised work to Molecular Systems Biology. We are now globally satisfied with the modifications made and we will be able to accept your paper for publication pending the following minor amendments.

- The title of the papers still refers to a causal relationship between changes in epitranscriptomics and translational regulation. We would suggest to modify the title to the following: "tRNA epitranscriptomics and biased codon usage are linked to proteome expression in *Plasmodium falciparum*".
- The datasets produced in this study should be deposited in a appropriate public database (see <http://msb.embopress.org/authorguide#dataavailability>). The accession numbers and database should be listed in a formal "Data availability" section (placed after Materials & Method) that follows the

model below (see also <http://msb.embopress.org/authorguide#dataavailability>):

Data and software availability

The datasets and computer code produced in this study are available in the following databases:

- RNA-Seq data: Gene Expression Omnibus GSE46843
(<https://www.ncbi.nlm.nih.gov/geo/query/acc.cgi?acc=GSE46843>)
- Chip-Seq data: Gene Expression Omnibus GSE46748
(<https://www.ncbi.nlm.nih.gov/geo/query/acc.cgi?acc=GSE46748>)
- Protein interaction AP-MS data: PRIDE PXD000208
(<http://www.ebi.ac.uk/pride/archive/projects/PXD000208>)
- Imaging dataset: Image Data Resource <https://doi.org/10.17867/10000101>
- Modeling computer scripts: GitHub
(<https://github.com/SysBioChalmers/GECKO/releases/tag/v1.0>)
- [data type]: [name of the resource] [accession number/identifier/doi] ([URL or identifiers.org/DATABASE:ACCESSION])

Corresponding Author Name: Peter Preiser
 Journal Submitted to: Molecular Systems Biology
 Manuscript Number: MSB-17-8009